# Research Progress on Extraction and Detection Technologies of Flavonoid Compounds in Foods

**DOI:** 10.3390/foods13040628

**Published:** 2024-02-19

**Authors:** Wen Li, Xiaoping Zhang, Shuanglong Wang, Xiaofei Gao, Xinglei Zhang

**Affiliations:** Jiangxi Key Laboratory for Mass Spectrometry and Instrumentation, East China University of Technology, Nanchang 330013, China; 2022120187@ecut.edu.cn (W.L.); 201660020@ecut.edu.cn (X.Z.); 201760075@ecut.edu.cn (S.W.); 201760059@ecut.edu.cn (X.G.)

**Keywords:** flavonoid, natural products, detection, extraction techniques

## Abstract

Flavonoid compounds have a variety of biological activities and play an essential role in preventing the occurrence of metabolic diseases. However, many structurally similar flavonoids are present in foods and are usually in low concentrations, which increases the difficulty of their isolation and identification. Therefore, developing and optimizing effective extraction and detection methods for extracting flavonoids from food is essential. In this review, we review the structure, classification, and chemical properties of flavonoids. The research progress on the extraction and detection of flavonoids in foods in recent years is comprehensively summarized, as is the application of mathematical models in optimizing experimental conditions. The results provide a theoretical basis and technical support for detecting and analyzing high-purity flavonoids in foods.

## 1. Introduction

Flavonoids are a large class of polyphenolic compounds commonly found in vegetables, fruits, cereals, tea, and Chinese herbal medicine. Most flavonoids exist in different modified forms, such as hydroxylation, methylation, acylation, and glycosylation, of which glycosylation is the most common form of modification of flavonoids [1]. The glycosylation of flavonoids, coupling flavonoid aglycones and glycosyl groups in conjugated form, can change their water solubility and stability and give them new biological activities and functions [2]. However, flavonoid glycosides are too water-soluble to diffuse across cell membranes. Therefore, the absorption of flavonoid glycosides in the human body generally requires the hydrolysis of their sugar group or active transport by specific enzymes [3,4]. Flavonoids have a basic C_6_–C_3_–C_6_ skeleton composed of two benzene rings (A and B) linked to each other by the central three carbons. Based on the oxidation degree of the central heterocycle, the saturation degree of the segment C_3_, and the insertion position of substituents, flavonoids can be divided into several categories, mainly classified as flavanones, flavones, isoflavones, flavonols, flavanols, and anthocyanins [5].

Each of these flavonoids is widely distributed in nature. The basic structure, classification, and food sources of flavonoids are summarized in Figure 1. Flavanones, also known as dihydroflavones, are characterized by the absence of double bonds in the 2, 3 positions of the pyranone ring and are isomers with Chalcones. They are commonly found in the vacuoles of plants such as citrus in the form of glycoside derivatives. Isoflavones are flavonoid compounds fused at the C3 position of B ring and C ring. They are phenolic substances with phytoestrogen activity and often exist in grains such as soybeans, black beans, broad beans, and nuts. Glycitein is an O-methylated isoflavone that makes up 5–10% of total isoflavones from soy foods [6]. Anthocyanins are plant pigments that render various pink, red, purple, and blue hues in flowers, vegetables, and fruits. Structurally, anthocyanins are anthocyanidins modified by sugars and acyl acids. Because of their bright color, they are commonly used as an additive in the food industry. Flavanols are broadly characterized by the absence of a double bond between positions 2 and 3 of the C ring and feature a hydroxyl group at position 3. Flavanols are commonly found in grapes, apples, plums, tea, cocoa, and beer. Catechins and epicatechins are the most common monomeric flavanols polymerized through C–C bonds to form proanthocyanidins. They are usually connected by C4–C8 or C4–C6 bonds and can be divided into monomeric, oligomeric, and oligomeric proanthocyanidins according to the degree of polymerization. Flavonols are characterized by several specific substitutions in the A and B rings, which are connected with a three-carbon chain. Structure–activity studies indicate that the location of the catechol B ring on the pyran C ring and the quantity and position of hydroxy groups on the catechol group of the B ring affect the antioxidant activity of flavonoids [7]. Compared with other flavonoids, flavonols contain more 3-OH groups. Quercetin is the most common flavonoid glycoside compound in fruits, vegetables, and other plant-derived products.

Flavonoids are now considered crucial components in various nutraceutical, pharmaceutical, and medicinal applications. This is because they have many health benefits, including heart-protection, anti-cancer, anti-diabetes, anti-aging, and neuroprotective effects [8,9,10,11,12]. Antioxidant ability is an important function of flavonoid compounds. Flavonoids can transfer hydrogens and electrons to reactive oxygen and nitrogen species (RONS), stabilizing them and giving rise to relatively stable flavonoid radicals. They can also effectively remove free radicals in the body through indirect ways, such as inhibiting enzymes related to the production of free radicals, chelating transition metal ions such as Fe^3+^ and Cu^2+^ that have the effect of inducing oxidation, and enhancing the antioxidant capacity of other nutrients [13]. Flavonoids have anti-inflammatory properties through different mechanisms, mainly by inhibiting transcription factors (such as NF-κB) to modulate protein kinases, thereby inhibiting the production of inflammatory cytokines and inflammatory mediators [14].

Humans cannot synthesize flavonoids themselves and can only obtain them from their diet. Due to the complex and diverse biological activities of flavonoids and their potential health benefits and functions, they have been the focus of research in food science. The concentration of flavonoids in the food substrate and the rate at which they are absorbed into the site of action are key factors affecting their bioavailability [15]. Previous research has shown that consuming flavonoid-rich foods can reduce the risk of some diet-related chronic diseases, but high doses of certain flavonoids may have adverse effects [16]. Therefore, extraction and detection technologies play a key role in the action of flavonoids used for human health. The sources of flavonoids are often complex, with various isomers and complex structures. The isolation and detection of high-purity flavonoids have always been a bottleneck in developing and utilizing flavonoids. At present, many advanced extraction, separation, and detection techniques have been maturely applied to flavonoids. Traditional extraction methods such as maceration, reflux, and Soxhlet extraction are widely used in the extraction of flavonoids from food because of their simple operation and low equipment requirements, but the extraction time is long, and the organic solvent consumption is large, resulting in low yield. Reducing the analysis time and environmental pollution and developing innovative and environmentally friendly alternatives have been the main goals of the work carried out in recent years. Based on the shortcomings of traditional extraction techniques, researchers have developed ultrasound-assisted extraction, microwave-assisted extraction, enzyme-assisted extraction, supercritical fluid extraction, and accelerated solvent extraction for the environmentally friendly and efficient extraction and separation of flavonoids. Regarding analysis, spectrometric and chromatography techniques are commonly used to detect flavonoids. UV–Vis, fluorescence, and near-infrared spectroscopy based on spectroscopy have the advantages of simple operation, fast analysis speed, and high sensitivity. However, a single spectral technique is generally not used to detect flavonoid compounds due to the lack of selectivity and sensitivity. Chromatographic techniques are more sensitive than spectroscopic techniques, and high-performance liquid chromatography (HPLC) is the most widely used method for analyzing flavonoids. It can be combined with various detectors for qualitative and quantitative analysis of known flavonoids.

In recent years, due to the advantages of no sample pretreatment, high efficiency, and high selectivity, direct mass spectrometry has attracted much attention, providing a new method for directly analyzing plant tissue samples. So far, most reviews have focused on the specific extraction methods of flavonoid compounds, but there are few summaries of analytical methods. In this paper, the extraction and analysis methods of flavonoid compounds in foods were reviewed comprehensively and systematically. The principle and characteristics of modern extraction methods were introduced in detail, and the advantages and disadvantages of various detection methods based on spectrum, chromatography, and mass spectrometry were summarized, providing a reference for developing efficient extraction and detection methods of flavonoids in foods.

## 2. Extraction Technologies of Flavonoids

### 2.1. Conventional Extraction Technologies

Solid–liquid extraction is the most commonly used method to extract flavonoids from natural products. Common solid–liquid extraction methods, including maceration, reflux, and Soxhlet extraction, are widely used because they do not require expensive equipment and complex extraction processes [17]. The type of solvent and its polarity have a significant impact on the extraction of various components [18]. The lipophilic parameter value (Log P) of a biological substance is an important parameter for selecting an extraction solvent. Flavonoids are mainly in a glycosidic form with a low Log P, so polar solvents, such as methanol, ethanol, acetone, and water, are usually considered when selecting solvents. Boeing et al. [19] used a solid–liquid extraction method to extract anthocyanins from mulberry, blackberry, and strawberry and compared the effects of different solvent combinations on extraction efficiency. The highest anthocyanin extraction content was obtained for all the berries analyzed using methanol/water/acetic acid (70/29.5/0.5, v/v/v). Khalili et al. [20] studied the effects of different solvents on total phenolic content (TPC), total flavonoid content (TFC), antioxidant activity, and antibacterial properties of extracts from red onion (Allium cepa). The active components of red onion were extracted in a range of increasingly polar solvents (ethyl acetate, n-butanol, ethanol, methanol, and water). The results showed that the solvent with a lower polarity yielded higher phenolic and flavonoid contents, and the extracts acquired by ethyl acetate had the highest TPC (423.95 ± 11.39 mg GAE/g of extract) and TFC (29.59 ± 1.71 mg QE/g of extract). Alwazeer et al. [21] used solid–liquid extraction to extract phenols, flavonoids, anthocyanins, and antioxidants from lemon peel. They found that incorporating hydrogen into all solvents can increase the extraction of all phytochemicals. The experimental results showed that the extraction yield of flavonoids in hydrogen-rich water, hydrogen-rich methanol, and hydrogen-rich ethanol increased by 20.59%, 30.33%, and 41.5% at 25 °C, respectively. This is because the standard reduction potential of 2H^+^/H_2_ involved in the biological electron transfer system equals −420 mV (at pH 7), showing that dissolved hydrogen possesses a possible thermodynamic reducing property to help maintain the redox homeostasis inside the cell and preserve phenolics and antioxidants from oxidative reactions. Compared with conventional organic solvents, the addition of H_2_ gas is a relatively cheap and practical method to improve the extraction efficiency of flavonoids from foods. In addition, liquid–liquid extraction with two different polar non-miscible solvents such as ethyl acetate/water can extract flavonoids very efficiently. This method does not require a large solvent volume and can be automated and applied to a large range of samples, including plants, foods, human biological fluids, or hair samples. For example, Xiao et al. [22] used a two-step liquid–liquid extraction method to extract ginkgo flavonoids, terpene lactones, and nimodipine from rat plasma. In the first step, nimodipine was extracted with hexane-ether (3:1, v/v), and then ginkgo flavones and terpene lactones were extracted with ethyl acetate. Combined with ultra-high-performance liquid chromatography/tandem mass spectrometry (UHPLC-MS/MS) analysis of the extracts, the results showed that the calibration curves of all analytes showed good linearity (R^2^ > 0.99), and the lower limits of quantification were 0.50–4.00 ng/mL.

Besides the solvents, the liquid–solid ratio, stirring speed, temperature, and extraction time are other key factors affecting extraction efficiency. Therefore, determining an optimal parameter helps achieve maximum extraction. Pereira et al. [23] used the maceration method to obtain bioactive substances (flavonoids, apigenin, and apigenin-7-glycoside) from Matricaria chamomilla inflorescences. The optimal extraction conditions were determined by response surface model and demand function as follows: the extraction was performed at 900 rpm for 1 h, the percentage of plant with respect to solvent was 36.8% (*w*/*w*), the ethanol solution was 74.7% (*w*/*w*), and the temperature was 69 °C. Under this set of conditions, the extract had total flavonoid content of 4.11 ± 0.07%. The main disadvantages of this method are the large volume of solvents, the long processing time, and the need for future purification. The Soxhlet extraction technique involves a smaller amount of solvent and a shorter extraction time. Ma et al. [24] used reflux extraction to obtain polyphenols (7.22 mg/g) from the shell of Pleioblastus amarus (Keng) and showed that the best extraction parameters were an ethanol concentration of 75%, a 20:1 liquid-to-solid ratio, and an extraction time of 2.1 h. Extraction with a Soxhlet device based on the maceration process can accelerate the extraction rate and improve the extraction efficiency to a certain extent. However, it is important to note that Soxhlet extraction is used only for extracts that contain thermostable flavonoids. For example, Daud et al. [25] used 70% ethanol (4 L) to extract antioxidants from Artocarpus heterophyllus wastes at boiling temperature. The results showed that the percent yield extraction of the Soxhlet extraction was only 12.1%. This is because the Soxhlet technique, which involves thermal reaction, could cause degradation of polar constituents (easily extractable by 70% EtOH), leading to low yields.

### 2.2. Advanced Extraction Technologies

Although the optimization of extraction conditions can greatly improve the extraction efficiency of flavonoid compounds, these methods require a large amount of solvent, time, and low purity, and it is difficult to achieve large-scale applications. Therefore, several new techniques have been used to eliminate the shortcomings of traditional methods, such as microwave-assisted extraction (MAE), ultrasonic-assisted extraction (UAE), enzyme-assisted extraction (EAE), accelerated solvent extraction (ASE), and supercritical fluid extraction (SFE).

Table 1 summarizes some of the most recent reports on extracting important flavonoids using modern techniques. MAE and UAE are the two most widely used technologies for extracting flavonoids from food and plants. Compared with the traditional extraction time of several hours or even days, MAE and UAE can effectively extract the target compounds in about 40 min. The ASE is time-consuming, but it needs to be extracted at a high temperature (50~200 °C) and high pressure (10.3~20.6 MPa), so it is unsuitable for thermosensitive compounds. In all extraction methods, ethanol is the most commonly used solvent, and higher yields have been obtained in the recovery of flavonoids. In addition, the extraction of flavonoids is affected by time, temperature, solvent concentration, solvent polarity, and other parameters, and each analysis method has different optimal extraction conditions for target products with different properties. According to different target molecules, different factors and their interactions are studied to determine the best experimental conditions that can achieve the maximum extraction efficiency.

#### 2.2.1. Microwave-Assisted Extraction

The microwave-assisted extraction (MAE) of flavonoids uses the energy of microwave radiation to accelerate the release of target compounds from solvents in plant materials [61]. Poureini et al. [26] used the MAE method to extract apigenin from parsley leaves. The results show that the extraction time, solvent use, and energy consumption of this method are significantly reduced compared with the traditional techniques, which proves the high efficiency of the modern extraction methods. Microwave radiation can affect the polarity and permittivity of the solvent, thus changing the solvent’s ability to dissolve flavonoids. Therefore, selecting a suitable solvent can improve the extraction efficiency of flavonoids. Sonar et al. [62] studied the effect of solvent selection on extraction efficiency when MAE was used to extract active marker compounds from *Aegle marmelos*. Ethanol, methanol, and various compositions of ethanol and water were used in the experiments. The results showed that methanol had the highest yield, but it was not selected because of its toxicity. With further study, the largest extraction rate was obtained in 80% (*v*/*v*) aqueous ethanol due to the increase in the polarity of ethanol by the addition of water, which eventually increases the mass transfer, increasing the extraction yield. Ihsanpuro et al. [63] used the MAE method to extract bioactive compounds from the pericarp and seeds of mangosteen and studied the effects of green solvents, including EtOH, EtOAc, and IPA, on the extraction rate. The results showed that the highest yield of mangosteen pericarp extract, which was obtained from extraction using EtOH solvent, was 14.74%, and the lowest yield was 8.56% with EtOAc solvent. The higher the proportion of EtOH and IPA in the single-phase solvent mixture, the higher the yield that is obtained. This is because the dielectric constant of EtOH and IPA is higher than that of EtOAc. It is further proved that water with a high dielectric constant can increase the yield value of the extract when using two-phase solvent.

The extraction of bioactive substances with MAE is affected by many parameters, including irradiation time, temperature, liquid–solid ratio, solvent concentration, and microwave power. Therefore, many researchers have considered these variables and their interactions using statistical and mathematical modeling techniques to determine the best extraction conditions to obtain the highest extraction efficiency. Response surface methods (RSM) are now commonly used to optimize extraction conditions. RSM is a complex mathematical method for evaluating the effects of many factors and their interactions on one or more response variables to optimize a particular experimental setup [64,65,66]. Alara et al. [67] used the response surface method to study the effect of the MAE variable on TFC and antioxidant activity of *Vernonia amygdalina* leaf. Under the optimal conditions of a irradiation time of 7 min, a microwave power of 416 W, a temperature of 100 °C, and a solid–liquid ratio of 0.10 g/mL, the highest TFC (87.05 ± 1.03 mg QE/g) was extracted, with a DPPH antioxidant capacity of 94.05 ± 1.03% (n = 3) and an ABTS antioxidant capacity of 95.93 ± 0.99% (n = 3). Compared with the Soxhlet extraction process, the optimized extract had a higher antioxidant activity and more chemical components. Kumar et al. [68] used the Box–Behnken design of the response surface method to study and optimized the MAE of phenols, antioxidant activity, and color density in black carrot residue. At the optimized conditions, the MAE yielded a higher color density (68.63 ± 5.40), polyphenolic content (264.9 ± 10.025 mg GAE/100 mL), TFC (1662.22 ± 47.3 mg QE/L), and AOC (13.14 ± 1.05 µmol TE/mL) in a short time as compared to other methods. Saifullah et al. [31] used MAE and RSM to optimize extraction parameters to maximize the extraction rate and antioxidant properties of phenolic compounds. Under optimal experimental conditions, MAE is eight times faster and requires six times less solvent than UAE and shaking water bath.

Overall, compared with traditional methods, MAE reduced the extraction time and solvent volume, but the condition of microwave heating may cause oxidation and degradation of target molecules. MAE is mainly limited to small polar molecules and is generally unsuitable for thermosensitive compounds.

#### 2.2.2. Ultrasound-Assisted Extraction

Ultrasonic-assisted extraction (UAE) is considered an environmentally friendly extraction technology as it can help reduce the extraction time and energy consumption compared to conventional extraction techniques [69].

The temperature at which the ultrasound-assisted extraction of flavonoids takes place is usually within the range of 30–60 °C, and most flavonoids can be extracted within 40 min to obtain the best extraction efficiency. Therefore, UAE is commonly used for the extraction of thermolabile and unstable compounds. Ultrasonic energy can accelerate the extraction of flavonoids. However, excessive ultrasonic energy may have an effect on active phytochemicals by forming free radicals [70]. Therefore, UAE also needs to optimize ultrasonic parameters compared with traditional extraction techniques. Garcia-Castello et al. [71] compared the efficiency of conventional solid–liquid extraction and UAE for flavonoid extraction from grapefruit peels. The effect of operating variables such as ethanol/water ratio, extraction temperature, and extraction time on the yield of phenolic compounds and antioxidant activity was evaluated according to the RSM approach. The result showed that UAE was more than 50% more efficient regarding the bioactives extraction yield. Giordano et al. [72] used the ultrasonic-assisted extraction of flavonoids from kiwi peel. The experimental conditions were optimized based on the response surface method, and the results show that the UAE was significantly affected by the independent variables of time, ultrasonic power, and ethanol concentration. The sonication of the sample at 94.4 W for 14.8 min, using 68.4% ethanol, resulted in a maximum of 1.51 ± 0.04 mg of flavonoids per g of extract. Park et al. [73] employed the Box–Behnken design (BBD) to optimize the UAE process for extracting antioxidants from Ruby S apple peel. Optimal conditions were established as 20 °C extraction temperature, 25.30 min extraction time, and 50% ethanol concentration. Under these optimal conditions, the highest content of antioxidants was extracted, among which the highest content of total flavonoids was 4.09 ± 0.05 mg NAE/g DW. Vo et al. [42] analyzed the effects of solid–liquid ratio (SLR), acetone concentration (AC), temperature, and time on the extraction of total phenol and flavonoid compounds from watermelon rind. The BBD model was employed to optimize the UAE conditions, and the optimal conditions were 1:30.50 SLR, 70.71% AC, 29.78 °C, and 10.65 min extraction time. The results provided a feasible and innovative process for the extraction of active components from watermelon rind. Singh et al. [43] used an artificial neural network (ANN) and particle swarm optimization (PSO) to optimize the experimental conditions for UAE to maximize antioxidant and antimicrobial activity from green coconut shells. The results showed that UAE was superior to MAE, and it was a feasible and successful method to extract phytochemicals from green coconut shells.

#### 2.2.3. Enzyme-Assisted Extraction

Compared with physical treatment, enzyme-assisted extraction can destroy cell walls more effectively and reduce the loss of active compounds. Thus, it was widely used in the extraction of many other active components of natural products. Enzyme-assisted extraction mainly depends on the ability of enzymes to hydrolyze cell wall components and disrupt cell walls’ structural complexity [74]. The most commonly used enzymes for the extraction of bioactive flavonoids are cellulases, α-amylase, protease, and pectinase. In the enzyme-aided extraction process, the operational conditions, such as the temperature of reaction, pH of the system, enzyme concentration, particle size of substrate, and time of extraction, are important. Due to the effect of temperature and pH on enzyme activity, the temperature for enzyme-assisted extraction of flavonoids is usually within 60 °C and the pH value is between 5 and 8. Elsayed et al. [48] used cellulase, protease, and their combination (1:1) to extract phenolic compounds (EAE) from maize male ears and optimized them using the central composite response surface method. Under optimum enzymatic conditions, the experimental TPC values were 9.78, 8.45, and 10.70 mg/g, respectively, which were significantly higher than that of the non-enzymatic control (6.75 mg/g). The results also showed that the concentration of cellulase and protease was the key factor affecting the extraction efficiency. The extraction amounts of phenolic compounds in corn tassel extracted using the enzyme-assisted method were increased by 41.62% (cellulase), 25.62% (protease), and 59.11% (cellulase-protease mixture 1:1). Compared with a single-enzyme-assisted extraction, more phenolic compounds were quantitatively identified in the mixed-enzyme-assisted extraction of corn tassel. Qadir et al. [47] used RSM and ANN to evaluate the effect of enzyme complexes on the recovery of phenolic substances from Capparis spinosa fruit extract. Under the selected parameters (time, pH, enzyme concentration, and temperature), the enzyme concentration greatly improved the extract yield. The extraction yield of 6.5% enzyme concentration (42%) was much higher than that of 0.5% enzyme concentration (29%).

According to literature reports, enzyme-assisted extraction has been combined with a variety of techniques for extracting flavonoids [50,75,76,77]. The advantages of various extraction methods can be complemented and the extraction rate can be improved by the comprehensive utilization of various extraction methods. Xu et al. [50] investigated enzyme-assisted ultrasound-microwave synergistic extraction (EAUMSE) of flavonoids from Chinese water chestnut peels. The yield of total flavonoids was 1.48% (*w*/*w*) at an enzymatic hydrolysis time of 2 h, hydrolysis temperature of 50 °C, enzyme concentration of 1.5%, ratio of cellulase to pectinase of 2:1 (*w*/*w*), and enzymatic hydrolysis of pH 5.0. The flavonoid yields from using EAUMSE were 26.50%, 22.31%, and 12.98% higher than from solvent extraction, UAE, and MAE, respectively. Van et al. [51] extracted bioactive compounds from citrus peels using the combined enzyme and ultrasound-assisted extraction (E-UAE) or ultrasound and enzyme-assisted extraction (U-EAE) technique and compared them with the extraction techniques using EAE and UAE. The E-UAE was found to be the most effective technique to obtain the extracts having the highest TPC, TFC, naringin, and hesperidin contents and antioxidant and antimicrobial activities.

#### 2.2.4. Accelerated Solvent Extraction

Accelerated solvent extraction (ASE), also known as pressurized liquid extraction (PLE), is a new technology to extract the target substance in solids with an organic solvent under the conditions of increasing temperature and pressure [78]. Compared with traditional solvent extraction, ASE has the advantages of high extraction efficiency, short extraction time, and lower consumption of organic solvents [79]. Only a few minutes can realize the effective extraction of flavonoid compounds. Cai et al. [36] investigated the extraction efficiency of anthocyanins from purple sweet potatoes using conventional extraction, UAE, and ASE. The results showed that temperature was the key factor affecting the extraction efficiency of anthocyanins. The extraction efficiency of anthocyanins using ASE was the highest, and the anthocyanin yield from ASE ranged from 142.46 to 252.34 mg/100 g DW. This is because ASE is a pressurized-liquid, short-time extraction method at high temperatures, so it is advantageous for extracting heat-unstable anthocyanins from purple sweet potato. Conventional extraction and UAE are extraction methods that take place at relatively low temperatures over a long period of time, resulting in low anthocyanin yields and high impurities of other phenolic substances.

ASE is affected by several factors, such as extraction temperature, extraction time, and solvent composition, depending on the target compounds. Nandasiri et al. [53] found that the recovery of phenolic and flavonoid compounds from canola meal depends on the type and polarity of the solvent extractant, extraction method, and temperature. Compared with pure solvent, 70% ethanol aqueous solution and methanol extractant are more efficient in extracting phenolic compounds using the ASE method. The highest yield of phenolic compounds was obtained with 70% methanol (20.72 ± 1.47 mg SAE/g DM) and 70% ethanol (24.71 ± 2.77 mg SAE/g DM) at a temperature of 180 °C. Supasatyankul et al. [55] used pressurized liquid extraction to extract phenols and flavonoids from the mung bean (*Vigna radiata* L.) seed coat. RSM was used to optimize the extraction process, and the optimal extraction conditions were 160 °C, 1300 psi, and 50% ethanol. Bebek Markovinović et al. [54] developed a green extraction approach using PLE to provide the highest yield of bioactive compounds from strawberry tree fruit. By optimizing the extraction PLE process parameters, it was found that temperatures of 120 °C, a static extraction time of 10 min, and two cycles gave the highest yield of all bioactive compounds, and the yield of TF was 24.29 mg 100 g^−1^.

#### 2.2.5. Supercritical Fluid Extraction

Supercritical fluid extraction (SFE) is a new type of extraction and separation technology that has developed rapidly and has been widely used recently. CO_2_, C_2_H_4_, C_2_H_6_, CH_4_O, C_2_H_6_O, and H_2_O can be used as supercritical solvents [80]. CO_2_ is the most commonly used supercritical fluid. Due to its low critical pressure and critical temperature, it can be separated at a lower temperature [81]. In addition, the critical density of CO_2_ is higher than that of other commonly used supercritical solvents, and it has a strong ability to dissolve organic matter and will not cause damage to heat-sensitive substances and active ingredients.

Temperature, pressure, solvent flow rate, and extraction time are important parameters affecting the extraction process. These parameters can be optimized to improve the yield of flavonoid extraction. Song et al. [56] extracted flavonoids from Xinjiang Jujube leaves (XJL) at a temperature of 52.52 °C, a pressure of 27.12 MPa, a time of 113.42 min, and cosolvent flow rate of 0.44 mL/min by SFE-CO_2_. Compared with conventional Soxhlet extraction (CSE) and UAE extracts, the SFE-CO_2_ [60] extracts had higher concentrations of total flavonoids and stronger antioxidant and antiproliferative activities. Buelvas-Puello et al. [58] used supercritical carbon dioxide extraction to extract phenolic compounds from mango (*Mangifera indica* L.) seed kernels. The results showed that the extraction efficiency was the highest at 21.0 MPa, 60 °C, and 15% EtOH, with a total yield of 11.8%, TPC 19.4 mg-eq AG g^−1^ extract, and TFC 3.8 mg-eq Q g^−1^ extract. Kamaruddin et al. [59] used a central-composite design (CCD) approach to study and optimize the operating conditions for the SFE-CO_2_ extraction of 6-gingerol, TPC, and TFC. Under the conditions of a pressure of 25 MPa, a temperature of 40 °C, and a particle size of 300 µm, 6-gingerol content (171.26 mg/g), TPC (17.84 GAE mg/g), and TFC (74.46 QE mg/g) were extracted from Bentong ginger. And the extraction efficiency is much higher than that of traditional Soxhlet extraction. Kaur et al. [60] used SC-CO_2_ technology to extract high-value phenolic compounds from rice husk and studied the effects of different SC-CO_2_ modifiers (i.e., ethanol and ethanol-water) on the extraction efficiency of phenolic compounds. The results showed that increasing the water content of the cosolvent to 50% (*v*/*v*) could significantly improve the extraction rate.

#### 2.2.6. Application of Deep Eutectic Solvents

Many current extraction methods require the consumption of organic solvents, the use of which involves many processing procedures and requires prolonged high-temperature conditions, which can lead to the ionization, hydrolysis, oxidation, and deactivation of flavonoids [56]. Due to the harmful effects and low extractability of conventional solvents, there are significant limitations in large-scale separations. Deep eutectic solvents (DESs) have excellent solubility compared with traditional organic solvents. The solubility of DES in compounds with different properties varies according to the combination of hydrogen bond donor (HBD) and hydrogen bond acceptor (HBA). This means that the solubility and extraction efficiency of the target compound can be improved by selecting the right components. At present, DESs have been combined with various extraction methods for the extraction of flavonoids.

Xu et al. [82] used DESs as the extraction solvent to extract and recover valuable flavonoids from citrus peel waste and evaluated the extraction effects of five different DESs. The results showed that the extraction rate of citrus flavonoids was linearly correlated with the polarity of the HBD. Under optimized conditions, choline chloride-levulinic acid-N-methyl urea provides a higher extraction yield of total flavonoids (65.82 mg/g) than the most effective conventional solvent (53.08 mg/g). Ali et al. [45] evaluated 11 different choline chloride-based DESs as solvent for extracting flavonoids from *Lycium barbarum* L. fruits, and the results showed that DES containing strong acid HBD formed from a 1:2 mixture of choline chloride and p-toluene sulfonic acid could achieve the highest extraction efficiency of flavonoids. All prepared DESs have a better affinity to flavonoid extraction than water, methanol, and ethanol from *Lycium barbarum* L. fruit powder. In addition, the study also found that the viscosity and polarity of DES to determine bioactive compounds are the main factors of extraction efficiency, and DES high viscosity reduced the bioactive compounds to the solvent diffusion rate. Vo et al. [83] used UAE combined with natural deep eutectic solvents (NADES) to extract phenolic compounds and flavonoids from black mulberry fruit and optimized the experimental conditions. The results indicated that NADES produced from choline chloride and lactic acid was a suitable solvent for extracting phenolics and flavonoids from black mulberry fruit and had a higher extraction yield than conventional solvent. Wang et al. [34] exploited a green and efficient DES-MAE approach to extract the seven main bioactive flavonoids from *Ribes mandshuricum* leaves. With choline chloride-lactic acid as the extraction solvent, the yield of seven target flavonoids reached 4.78, 2.57, 1.25, 1.15, 0.34, 0.32, and 0.093 mg/g DW, respectively, under the optimal experimental conditions.

### 2.3. Overview of Research Progress on Flavonoid Extraction

Research on flavonoid-extraction technology can greatly promote the discovery of new compounds, as well as the development of food analysis and drug research. Recent research has focused on environmentally friendly sample-extraction techniques, progressing from traditional extraction methods (maceration, reflux, and Soxhlet extraction) to developing green, modern extraction techniques (MAE, UAE, EAE, ASE, and SFE).

Although traditional techniques do not require special equipment, they are gradually being replaced by other methods due to their long extraction times, low extraction selectivity, and low efficiency. Modern extraction techniques are the most commonly used method at present, especially combined with deep eutectic solvents, effectively solving the harmful effects of traditional organic solvents and low extraction rates. It is often difficult to choose the most suitable extraction method for different target molecules. In addition to the properties of the target compounds, many factors such as solvent selection, temperature, extraction time, liquid–solid ratio, particle size, and pH value will affect the extraction efficiency of bioactive molecular compounds. Because of this problem, mathematical statistical models such as RSM and ANN are usually combined with machine learning algorithms to study different factors and their interactions to determine the best experimental conditions and achieve maximum extraction efficiency.

Although these techniques are relatively mature and have been widely used in extracting flavonoid compounds from different foods and plants, most of the extraction techniques only focus on the extraction yield of flavonoid compounds and ignore the purity problem. In the subsequent analysis, further purification steps are often required, and due to the complex chemical composition and low content of monomer components in the extract, there are still problems such as low efficiency and long time consumption in practical applications. Therefore, in the future research process, it should be developed in the direction of more efficient and more suitable for large-scale production, and researchers should try to combine the existing extraction technology and separation technology to establish a continuous extraction and separation automation equipment, which will greatly help the further development and utilization of natural flavonoid compounds.

## 3. Detection Technologies of Flavonoids

Spectroscopic and chromatographic techniques are the most commonly used methods for detecting flavonoids. Spectroscopy-based techniques such as ultraviolet, fluorescence, and near-infrared spectroscopy have the advantages of simple operation, fast analysis, and high sensitivity. Chromatographic techniques can be selectively used to determine single flavonoids and are more sensitive than spectroscopic techniques. Usually, the most used analytical method for flavonoid determination is carried out on HPLC because of its flexibility. The diversity of stationary, mobile, and coupled detectors can meet the requirements for the isolation and determination of most flavonoids from all sources. It should be noted that chromatographic techniques have limitations in the determination of crude extracts. Samples are usually cleaned before analysis; such operation may remove some compounds of interest. Therefore, analytical techniques without sample pretreatment have attracted much attention recently. Direct mass spectrometry provides a new method for the direct analysis of plant tissue samples, which can quickly obtain specific gaseous ions in complex substrate samples without sample pretreatment and greatly improve extraction efficiency. The advantages and disadvantages of these techniques for flavonoid determination are summarized in Table 2.

### 3.1. Spectrometric Techniques

#### 3.1.1. UV–Vis Spectrophotometry

Ultraviolet–visible (UV–Vis) spectrophotometry is a convenient, fast, and effective technique that uses the absorption spectra of compounds to qualitatively and quantitatively analyze compounds that absorb energy in the ultraviolet or visible region and undergo electron energy transitions [84]. Paula et al. [85] established a new analytical methodology for simultaneously measuring the total content of three classes of phenolic compounds (hydroxybenzoic acids, hydroxycinamic acids, and flavonoids) and total phenolic content in propolis extract using a single UV–Vis spectrum. Soylak et al. [86] developed a method for separating and enriching quercetin in sample solution media via amine liquid phase microextraction and determining quercetin using a UV–Vis spectrophotometer. As an extraction solvent, N,N-dimethyl-n-octylamine has been used, and the quercetin concentration in the extraction phase was determined via UV–Vis spectrophotometry at 382.5 nm. The detection limit (LOD) and the quantitation limit (LOQ) values for quercetin in the sample solution were calculated as 0.07 μg·mL^−1^ and 0.24 μg·mL^−1^, respectively. This detection method is simple to operate; does not require special, expensive, or specific equipment; and has been successfully applied to food samples such as spinach, green pepper, and red onion.

UV–vis spectrophotometry for the determination of flavonoids has the advantages of rapid analysis, low cost, and simple technology. However, compared with other methods such as HPLC, GC-MS, and LC-MS, the detection limit of ultraviolet spectrophotometry is lower, and the extracted trace compounds often need to be enriched before detection. Therefore, many researchers have used a variety of technologies to detect the content and characterize the structure of flavonoids. Khakimov et al. [87] combine ultraviolet–visible (UV–Vis) as a low-cost, non-destructive fingerprinting technique with the high-throughput and separation capabilities of ultra-high-performance liquid chromatography–mass spectrometry (UHPLC-MS) and gas chromatography–mass spectrometry (GC-MS) to show potential for analyzing non-volatile components of a variety of fruits. Kreps et al. [88] used electron paramagnetic resonance (EPR) and ultraviolet–visible (UV–Vis) to analyze the antioxidant activity, TPC, and TFC in extracts of ethanol, methanol, and acetone from sea buckthorn juice. The result showed that a 70% ethanol extract of sea buckthorn juice had an average of 1.3 and 1.6 times greater TPC and TFC values, respectively, than other extracts. Hu et al. [89] isolated three new flavonoid glycosides and eight known compounds from the aerial parts of *Allium sativum* and identified the structure of these compounds in the extracts using ultraviolet–visible (UV–Vis), infrared (IR), nuclear magnetic resonance (NMR), and high-resolution electrospray ionization–mass spectrometry (HR-ESI-MS). The results showed that nine components were active, which revealed the pharmacodynamic material basis of the aerial parts of *A. sativum* on platelet aggregation.

#### 3.1.2. Fluorescence Spectroscopy

Fluorescence is a sensitive and selective analytical technique, usually 2–3 orders of magnitude higher than a spectrophotometer [90]. Turturică et al. [91] studied the effect of heat treatment on the degradation of polyphenol compounds in sweet cherry extract within the range of 70–120 °C using fluorescence spectroscopy and spectrophotometry. Shan et al. [92] used synchronous fluorescence spectra coupled with chemometrics to determine the flavonoid content of tea brews rapidly and non-destructively. By using the continuous projection algorithm and interval partial least squares regression to extract the wavelength of the information, the optimal interval partial least squares regression model is obtained. Catechin, a kind of flavanol widely present in tea, has a strong antioxidant capacity and can effectively prevent cancer. Du et al. [90] combined the standard addition method and fluorescence spectroscopy to determine the catechin content in black tea with different concentrations at the optimal excitation and emission wavelength. The results indicated a linear relationship between the obtained concentration and fluorescence intensity, where the R values were all greater than 0.99 and the LOQ was 0.02 μg·mL^−1^. Due to the quenching effect of a high concentration of catechins, the low concentration of diluted tea liquid was used to determine the fluorescence spectrum, further improving the determination limit. This method is sensitive, rapid, and low in cost, and it has been successfully applied to the determination of catechins in black tea from different regions.

#### 3.1.3. Nuclear Magnetic Resonance

Nuclear magnetic resonance (NMR) refers to the phenomenon that the nuclear energy level of an atom undergoes a splitting under the action of a strong magnetic field and that the transition of the energy level occurs under specific radio frequency radiation [93]. Different atoms have their own resonance signatures, and NMR spectroscopy can identify the number, type, and relative position of certain atoms in a molecule by capturing their resonance characteristics. Among the commonly used spectroscopic methods for flavonoid compounds analysis, NMR spectroscopy shows great advantages because of its high level of structural information and the fact that the sample is not destroyed [94]. NMR technology covers a variety of nuclei, such as ^1^H, ^13^C, ^19^F, ^17^O, ^31^P, etc. [95]. ^1^H-NMR is widely used to identify the molecular structure of flavonoids because of its high sensitivity. Lund et al. [96] quantized four flavonoids (naringenin, hyperoside, rutin, and vitexin-2″-O-rhamnoside) and chlorogenic acid in leaf extracts of four Crataegus species by ^1^H-NMR, revealing chemical differences between the four biological samples. Bationo et al. [97] isolated three flavonoids from the leaves of the Burkina Faso species and identified the chemical structures of several flavonoids using ^1^H-NMR. Kontogianni et al. [98] used 1-D ^1^H NMR, 1-D total correlation spectroscopy, 2-D ^1^H–^13^C heteronuclear single quantum coherence, and ^1^H–^13^C heteronuclear multiple bond correlation nuclear magnetic resonance techniques for the simultaneous identification and quantification of artemisinin and five of its analogs, along with five flavonoids, an aromatic ketone, and camphor (in total, 13 compounds) in crude diethyl ether Artemisia annua. These methods do not require the laborious separation of individual analytes or extensive sample-preparation procedures, and the results are validated in terms of precision, linearity, and detection limits. Yi et al. [99] determined the ^1^H and ^13^C NMR chemical shifts and nuclear magnetic shielding parameters of daidzein and puerarin using NMR. The measurements were supported by theoretical considerations within the density functional theory (DFT) methodology. Liu et al. [100] isolated four new flavonoids and fourteen known compounds from the aerial part of Bupleurum chinense DC and determined their structures using NMR spectroscopy and high-resolution mass spectrometric analysis.

#### 3.1.4. Near-Infrared Spectroscopy

Near-infrared spectroscopy (NIR) is an electromagnetic wave between the visible light (Vis) and mid-infrared region (MIR), with a wavelength range of 800–2500 nm. By studying the frequency doubling and frequency combination absorption of near-infrared light on the vibration of hydrogen-containing groups X-H (X = C, N, O), one can analyze the organic composition and structure information can be analyzed and determine the component content [101]. Near-infrared spectroscopy (NIR) is a fast and non-destructive real-time online detection method that does not require pre-treating the sample and does not damage the tested sample [102]. Betances-Salcedo et al. [103] developed a quick method to quantify the composition of flavones, flavonols, flavanones, and dihydroflavonols in propolis using NIR with a reflectance fiber-optic probe applied directly to the ground-up sample of propolis. Escuredo et al. [104] used the same method to accurately predict potato tubers’ phenol content, flavonoid content, antioxidant capacity, dry matter, texture, and soluble solid content. Ye et al. [105] used NIR to achieve the rapid determination of total components (flavonoids, phenols, and organic acids) and monomer components (chlorogenic acid, hyperoside, and isoquercitrin) in Shanzha.

The analysis and interpretation of NIR spectra require appropriate stoichiometric tools, which can be combined with mathematical and statistical methods to extract information from multivariate spectral data, establish calibration models, and eliminate invalid and irrelevant information in NIR data using signal processing, pre-processing, curve fitting, pattern recognition, training, and verification sets, so as to realize the practical application of NIR spectra [106]. Ze et al. [107] used the improved weighted partial least squares (PLS) algorithm combined with NIR to construct a rapid calibration model to determine four main components of tea polyphenols, tea polysaccharides, total flavonoids, and theanine. It reveals the potential of NIR spectroscopy combined with multivariate calibration analysis applied to Pu’er tea quality assessment. Rouxinol et al. [108] used a portable near-infrared spectrometer to simultaneously evaluate six different quality parameters (soluble solids content, titratable acidity, total phenolic compounds, total anthocyanins, total flavonoids, and total tannin contents) of four different varieties of whole grapes and established a model to quantify important quality attributes of wine grapes using the partial least squares regression (PLSR) algorithm. The results showed that the coefficients of all the developed prediction models exceeded 81%, except that the R^2^ of total flavonoids and total phenols were 72% and 71%, respectively. The Residual Prediction Deviation (RPD) values of soluble solids content, titratable acidity, total anthocyanins, and total tannins in the prediction models were all greater than 2.3. Amanah et al. [109] based on Fourier transform near-infrared (FT-NIR) and Fourier transform infrared spectroscopy (FT-IR), combined with the PLS model, realized non-destructive prediction of isoflavones and oligosaccharides in intact soybean samples. The results show that the performance of the optimal prediction model of FT-NIR is acceptable (R^2^p: 0.80 and 0.72), which is slightly better than that of the model based on FT-IR data (R^2^p: 0.73 and 0.70). This method does not require complex sample pretreatment and is a promising strategy for non-destructive chemical-composition assessment of soybeans.

The PLS model is a prediction analysis model of full wavelengths and may include collinear, redundant, and irrelevant variables, which will affect the accuracy and stability of the built prediction model [106,110]. Haruna et al. [111] combined NIR spectroscopy with an efficient variable selection algorithm for the quantitative prediction of TFC and TPC in raw peanut seeds. The synergy interval (Si)-PLS model is a variable selection algorithm to collect and merge valid spectral intervals from spectral data. Competitive adaptive reweighted sampling (CARS) models are used to select variables to create high-performance predictive models combined with PLS. Therefore, the comprehensive application of the synergy interval coupled competitive adaptive reweighted sampling-partial least squares (Si-CARS-PLS) can be used to obtain the most important information for predicting the TFC and TPC in raw peanut seeds. Based on the correlation coefficients of prediction (Rp), root mean square error of prediction (RMSEP), and RPD, the performance of the model was evaluated. The results showed the Si-CARS-PLS yielded optimal performance, Rp = 0.9137, RPD = 2.4 for TFC and Rp = 0.9042, RPD = 2.3 for TPC. The combination of variable selection algorithms boosted the model performance with better accuracy and higher stability.

NIR spectroscopy generally collects spectral information from small sampling points and cannot obtain the spectra of the entire sampled region. Hyperspectral imaging (his) combines the advantages of spectroscopy and imaging and can obtain one-dimensional spectral information and two-dimensional spatial information of samples at the same time. Compared with the NIR spectrum obtained from a single point, HSI can obtain the spectrum of each pixel in the image at a specific wavelength or spectral region, making it feasible to predict the physical characteristics and chemical composition of a single pixel and obtain more representative spectral information [112]. Wang et al. [113] developed a stoichiometric-assisted HSI method to predict nutrient composition, including pectin polysaccharides (PPS), reducing sugars (RS), total flavonoids (TF), and total phenols (TP), enabling geographical origin differentiation of red raspberry. Zhang et al. [114] determined the total anthocyanins, total flavonoids, and total phenols in dry black goji berries using near-infrared hyperspectral imaging. They successfully predicted the chemical composition of black wolfberry by using deep convolutional neural network (CNN) modeling, indicating that deep learning has great potential as a modeling and feature extraction method for the chemical composition determination of NIR-HSI.

### 3.2. Chromatographic Techniques

#### 3.2.1. Supercritical Fluid Chromatography

Supercritical fluid chromatography (SFC) is a chromatographic separation technology with supercritical fluid as a mobile-phase and solid adsorbent or polymer bonded to the carrier as a stationary phase [115]. CO_2_ is inert, non-toxic, non-flammable, and easy to prepare, and its tunable solvent strength can be varied by varying the pressure–temperature conditions. Due to its low polarity, CO_2_ is unsuitable for working with polar analytes, but it can be miscible with liquid organic solvents with a wide range of polarity in a wide range of pressure and temperature, so SFC usually uses compressed carbon dioxide (CO_2_) mixed with a certain amount of organic solvent as the mobile phase [116]. Modifiers play an important role in the SFC separation since they can generate interactions with the analytes, thus increasing their affinity for the mobile phase and competing with the analytes for the hydrogen bonding donor and acceptor sites of the stationary phase [117]. Methanol is the most polar alcohol and leads to the higher polarity of the bulk mobile phase, which results in lower retention of most analytes [118]. Thus, methanol is the most commonly used mobile phase modifier in SFC.

Huang et al. [119] performed the baseline separation of 12 flavonoids on a Zorbax RX-SIL column within 18 min using gradient elution. They found that a methanol solution of 0.1% phosphoric acid was the most suitable polar mobile-phase component for the separation of flavonoids. Compared to a reversed-phase liquid chromatography (RPLC) method, the SFC method could increase the separation speed of flavonoids by about three times while maintaining good peak shape and comparable peak efficiency. Jiang et al. [117] developed a rapid and efficient supercritical fluid chromatography coupled with a diode array detection (SFC-DAD) method for the simultaneous quantification of flavonoids in Citri Reticulatae Pericarpium (CPR). Six flavonoids (tangeretin, 3,5,6,7,8,3′,4′-heptamethoxyflavone, nobiletin sinensetin, hesperidin, and didymin) were isolated from CRP within 10 min using supercritical CO_2_ and methanol as the mobile phase and Zorbax RX-SIL, a column temperature of 45 °C, a back pressure of 95 bar, and a flow rate 0.8 mL/min. The LODs were between 0.35 and 1.83 µg/mL, and the LOQs were from 0.94 to 4.76 µg/mL. Compared with the HPLC method, the SFC approach showed a shorter analysis time and a lower consumption of organic solvents. Li et al. [120] developed an efficient ultra-high-performance supercritical fluid chromatography (UHPSFC) method for the isolation and simultaneous determination of four target flavonoids (tangeretin, nobiletin, hesperetin and naringenin) from citrus samples. Compared with other separation and purification technologies, SFC technology overcomes the disadvantages of waste organic solvents, large pollution, and a long time. It has the advantages of a low cost, environmental friendliness, and high separation efficiency. It is a green separation and purification technology.

#### 3.2.2. High-Performance Liquid Chromatography

High-performance liquid chromatography (HPLC) is a technique that separates the mixture in the mobile phase through the chromatographic column. It can be used with different detectors that can also characterize the sample. The latter is separated by the chromatographic column and enters the detector with the mobile phase. Then, the detector converts the physical or chemical signal of the sample into an electrical signal to obtain a chromatogram of the different sample components. The identification of flavonoids is usually carried out by comparing the retention time obtained with that of a real sample and by analyzing their characteristics collected by a detector [121]. A diode array detector (DAD), ultraviolet absorption detector (UVD), and fluorescence detector (FLD) are common detectors in HPLC [122]. DAD is also known as a photodiode array detector (PDAD), a detector that uses the photodiode array as the detection element, conducts multi-channel parallel detection, and obtains three-dimensional spectral data of time, wavelength, and absorbance at the same time. The DAD completes the full wavelength scanning at the same time, which will cause a loss of accuracy, so the detection accuracy of the UVD is higher than that of the DAD.

Traditionally, HPLC combined with DAD was often used to analyze the flavonoid composition in foods. For example, Mesquita et al. [123] developed the HPLC-DAD method for the determination of the phenolic compounds profile of orange juice and combined with principal component analysis (PCA) to realize the distinction between traditional juice and organic juice. Jang et al. [124] devised a method for isolating flavonoid isomers from common buckwheat sprouts using HPLC-DAD. The optimum analysis conditions were determined as column temperature 40 °C with 0.1% (*v*/*v*) acidic water and acetonitrile as mobile phases and at a flow rate of 1 mL min^−1^. The minimal detection limits for the five components measured ranged from 0.09 to 0.42 µg mL^−1^, with recoveries ranging from 96.67% to 103.60%. The method is accurate and reliable and can be used to simultaneously analyze flavone and flavonol isomers in foods. Fu et al. [125] developed an accurate HPLC-DAD method to evaluate the quality of Artemisia annul and applied it to simultaneously quantify five flavonoids (rutin, cynaroside, isorhamnetin, isorhamnetin, chrysosplenol D, and casticin). The HPLC-DAD technique was employed by Hyeon et al. [126] to determine ten flavonoids (rutin, narirutin, naringin, hesperidin, neohesperidin, quercetin, naringenin, hesperidin, nobiletin, and tangeretin) in five types of citrus fruit species. The samples were separated using a flow rate of 0.8 mL/min, a column temperature of 40 °C, a mobile phase buffer of 0.5% acetic acid, and a detection wavelength of 278 nm. The results showed that the method had good linearity (R^2^ ≥ 0.9997), precision (daytime < 0.599%, intra-day < 0.055%), and accuracy (recovery 92.30–108.80%), which provided a practical and reliable method for citrus quality assessment. Yuan et al. [127] used HPLC and the UV method to determine the content of total flavonoids in *Abrus precatorius* leaves and compared their differences. The content of total flavonoids determined using the optimized process was 69.0 mg/g by HPLC and 41.7 mg/g by UV. The results showed that the UV method had difficulty quantifying the content of total flavonoids in Chinese medicine accurately. This is mainly related to the diversity of the parent nucleus structure of flavonoids in traditional Chinese medicine, which leads to quantitative errors in the determination of the total flavonoid content. Although the UV method is simple and rapid, it has difficulty reflecting the total flavonoid content accurately. Kim et al. [128] used HPLC-UV/VIS and HPLC-ESI-MS/MS to reveal the changes in flavonoids (rutin, quercetin and quercitrin) in the stems, leaves, and flowers of common buckwheat (CB) and Tartary buckwheat (TB) during growth. They identified a new flavonoid (quercitrin) in CB flowers. Compared with spectral analysis technology, HPLC has the advantages of a wide detection range, a low detection limit, and high sensitivity, but it is still limited in the detection of food flavonoid compounds due to the high cost of detection equipment and the long detection time of a single sample.

Flavanols, as a subclass of flavonoid compounds, including (−)-epicatechin and (+)-catechin, and their related oligomers, have received much attention in recent years due to their important role in promoting health. Flavanols and proanthocyanidins have various stereoisomeric forms. In addition to monomer flavanols and proanthocyanidin dimers, there are few analytical standards, and it is difficult to isolate and purify them [129]. Currently, most quantitative methods of flavanols in food rely on chromatographic technology and fluorescence detection. Bussy et al. [130] designed a method combining HPLC and FLD to quantify flavanols and proanthocyanidins in the range of 2 to 500 mg g^−1^. The results showed that the method had high precision (%RSD 0.2 to 1.9%) and accuracy (100.7 to 102.9%) and was successfully used to evaluate flavanols and proanthocyanidins in different cocoa products. Vidal-Casanella et al. [131] developed a reversed-phase high-performance liquid chromatography method with HPLC-UV/Vis and FLD to determine the content of major flavanols. The results showed that the HPLC-FLD method has better selectivity and sensitivity, and the detection limit is 0.1 mg L^−1^. In addition, hydrophilic interaction liquid chromatography (HILIC) can also obtain good separation of oligomers. Vidal-Casanella et al. [132] determined flavanols and related compounds in nutritional supplements based on HILIC and FLD techniques. The response surface method was used to optimize HILIC separation and achieve a good degree of separation of the main components. The difference in the composition characteristics of dietary supplements was revealed by combining the regression algorithm.

#### 3.2.3. Ultra-Performance Liquid Chromatography

Ultra-performance liquid chromatography (UPLC) is a derivative of HPLC and is superior to HPLC in many ways. UPLC uses a particle size of less than 2 µm, and due to the small particle size, the diffusion path between the stationary phase and the analyte is shorter, showing significant improvements in analytical speed, resolution, and sensitivity [133]. The system can operate at higher pressures, and the mobile phase can operate at higher linear speeds [134]. However, samples need to be cleaned before analysis, and this cleaning step may cause some target compounds to fall below the detection limit. Because of its key advantages such as high throughput and separation capacity, UPLC is widely used with various detectors for the discrimination of food origin and quality assessment.

Zhao et al. [135] devised a technique for determining the flavonoids in the peel and pulp of four citrus fruits using UPLC-PDA. Sixteen major flavonoids were identified using the C18 column with water/formic acid (99.99:0.01, *v*/*v*) (A) and methanol (B) as the mobile phases at a flow rate of 0.4 mL/min and temperature of 40 °C. Combined with PCA, the key compounds leading to the differences between different citrus varieties and fruit samples of the same variety were identified, providing a more advanced and feasible quality analysis method for the evaluation of citrus fruit quality. The LOD and LOQ of the method were less than 0.7167 and 1.7917 μg/mL, respectively, and showed significant advantages in terms of separation speed, with the identification of 9 min per sample, compared to 50 min per sample in previous studies to identify 16 flavonoids in the peel and pulp of citrus fruits using HPLC-DAD method [136]. Zhang et al. [137] used the C18 chromatographic column with a mixture of formic acid in water (0.1%) and acetonitrile as mobile phase A and B, respectively, to separate flavonoids from asparagus at a column temperature of 40 °C. Electrospray mass spectrometry ESI-MS was used to scan in negative ion mode, and 12 flavonoids were identified in asparagus tips and shoots for quantitative analysis. Otify et al. [138] used electrospray ionization-quadrupole time of flight mass spectrometry (UHPLC-qTOF-MS)-assisted molecular network to monitor 206 bioactive secondary metabolites in tomatoes, 30 of which were reported for the first time. Marcillo-Parra et al. [139] used HPLC-UV/VIS and UPLC-PDA to reveal differences in the content of bioactive compounds (e.g., polyphenols, flavonoids, carotenoids) and antioxidant activity in freeze-dried peel samples of different mango varieties.

### 3.3. Mass Spectrometry

#### 3.3.1. Liquid Chromatography–Mass Spectrometry

Due to the diversity and complexity of flavonoids, the effective isolation and precise quantification of some low concentrations of flavonoids using high-performance liquid chromatography and common detectors is challenging. Mass spectrometry (MS) is one of the most versatile and sensitive instrumental methods currently used for the structural characterization of plant secondary metabolic mixtures isolated from biological materials [140]. Mass spectrometry is an analytical method that is performed by measuring the mass–charge ratio of the ions in the sample. The sample to be tested is ionized, and the ions are separated according to the mass–charge ratio (*m*/*z*) according to the different motion behavior of different ions in the electric field or magnetic field to obtain the mass spectrum. Finally, the qualitative and quantitative results of the sample can be obtained through the mass spectrum of the sample and related information.

Compared to spectroscopy and chromatography, mass spectrometry is a more specific and highly selective detection technique that provides the molecular weight of molecular ions, which helps in the initial identification of flavonoids. Liquid chromatography–mass spectrometry (LC-MS) has become one of the most commonly used methods for the qualitative and quantitative analysis of flavonoids in nature. The combination of liquid chromatography (LC) and tandem mass spectrometry (MS/MS) has greatly facilitated the identification and quantification of flavonoid metabolites by combining the separation power of LC with the high sensitivity and selectivity of MS/MS. Wang et al. [141] extracted and determined major organic acids and flavonoids in Honeysuckle using LC-MS/MS. The results showed that the method exhibits good linearity. Li et al. [142] analyzed the difference in flavonoid metabolites in Tartary buckwheat and common buckwheat leaves by using UPLC-ESI-MS/MS combined with clustering analysis, PCA, and orthogonal signal correction and partial least squares-discriminant analysis (OPLS-DA), providing the theoretical basis for the future development and utilization of Tartary buckwheat leaves.

Mou et al. [143] carried out metabolite analysis based on ultra-high-performance liquid chromatography combined with electrospray ionization and mass spectrometry (UPLC-ESI-MS/MS) on four tissues of different citrus strains and detected 80 flavonoids. A total of 138 quantitative trait loci (QTLs) of 57 flavonoids were identified in these four tissues. The results provided a basis for characterizing the biosynthetic pathway of citrus flavonoids. Li et al. [144] used LC-MS/MS to detect the types and contents of flavonoids in Camellia oleifera honey and nine other monofloral honeys. The analytes were separated through gradient elution using water (0.05%, *v*/*v* formic acid) (A) and acetonitrile (0.05% formic acid, *v*/*v*) (B) at a flow rate of 0.35 mL/min on a C18 column. Combined with ESI-MS, the results showed that 54 flavonoids were detected in Camellia oleifera honey, which was higher than in the other nine other monofloral honeys. By constructing a PLS-DA model, they identified the distinct flavonoid marker in Camellia oleifera honey as kaempferitrin. This method provides a reliable and novel strategy for identifying the authenticity of Camellia oleifera honey.

Traditional mass spectrometry techniques, such as LC-MS, are time-consuming and have difficulty meeting the requirements of rapid analysis, as multi-step sample pretreatment must be performed before sample analysis, and long chromatographic run times are required to separate multiple components. Classical mass spectrometry ionization techniques include electrospray ionization (ESI), atmospheric pressure chemical ionization (APCI), substrate-assisted laser desorption Ionization (MALDI), and electron bombardment ionization (EI). However, these technologies belong to closed ionization, and the analysis of the sample needs to be closed in a certain pipeline or a certain vacuum environment and needs to transform the sample into different material forms.

In recent years, various atmospheric-pressure direct mass spectrometry techniques have been developed, such as desorption electrospray ionization–mass spectrometry (DESI-MS) [145], extractive electrospray ionization–mass spectrometry (EESI-MS) [146], and internal extractive electrospray ionization–mass spectrometry (iEESI-MS) [147], and direct analysis in real-time mass spectrometry (DART-MS) [148] provides a new method for the direct analysis of plant tissue samples by rapidly obtaining gaseous ions of specific components in complex matrix samples without sample pretreatment.

#### 3.3.2. Extractive Electrospray Ionization–Mass Spectrometry

Extractive electrospray ionization (EESI) is an environmental ionization technique in which an analyte of interest is extracted from a solution or gas sample and then ionized by the primary ions produced by the electrospray for subsequent mass spectrometry analysis [149]. EESI-MS is widely used to study the metabolic composition and change characteristics of plants. Liu et al. [150] detected the metabolites of citrus limon leaves after Asian citrus psyllid (ACP) treatment using EESI-MS and HPLC, and both EESI-MS and HPLC detected a significant increase in the contents of epicatechin and caffeic acid with the addition time of ACP. This study provides a new theoretical basis for the biochemical mechanism of ACP and its host plants. Xue et al. [151] used EESI-MS to analyze the metabolites in uninfected and Huanglongbing (HLB)-infected Newhall navel orange leaves. The mass spectrum of ethanol extract from Newhall navel orange leaves was obtained under the negative-ion mode, and 24 compounds, including flavonoids, were identified through comparison with the database. Based on the analysis results and multivariate analysis, Newhall navel orange leaves without infection and HLB infection were effectively distinguished. This method avoids the complicated process of sample metabolite separation, avoids the loss of information in the process, and can obtain more comprehensive metabolic information, which has great application potential in the early detection of HLB. Gao et al. [152] first established a simple and effective extractive electrospray ionization high-resolution mass spectrometry (EESI-HRMS) method to rapidly detect polyphenols and amino acids in three citrus species. Due to the high temperature, the substances to be measured may be destroyed. Therefore, the signal intensity of galangin (*m*/*z* 269.0444) and hesperetin (*m*/*z* 301.0707) was used to optimize the extraction agent and capillary temperature, and 80% methanol solution and 250 °C temperature were determined as the best experimental conditions. Combined with PCA analysis, major mass spectrum signals that help distinguish nectar from honey were identified, including *m*/*z* 179 (caffeic acid), *m*/*z* 269 (genistein), and *m*/*z* 301 (hesperidin) in qualitative compounds. The results of this study provide a reference for the identification of potential markers of citrus honey and the traceability of different varieties of honey. Light quality is a key factor affecting the growth, morphogenesis, physiological metabolism, and substance accumulation in plants [153]. Optimal light quality can improve the accumulation of metabolites in plants. Wu et al. [154] used neutral desorption–extractive electrospray ionization–mass spectrometry (ND-EESI-MS) to rapidly identify 20 compounds in raw leaves of Anoectochilus roxburghii within 20 s without any sample pretreatment. The results showed that the red-blue light increased the leaf area and enhanced the activity of antioxidant enzymes; increased the content of isoleucine, histidine, serine, and arginine in phenylpropanoid metabolism; and promoted promoting the accumulation of amino acids, kinsenoside, and caffeic acid, ferulic acid, quercetin, kaempferol, and rutin in the phenylpropane pathway.

#### 3.3.3. Direct Analysis in Real-Time Mass Spectrometry

Real-time direct analysis (DART) is a fast resolution and ionization technique. The principle is that under conditions of atmospheric pressure, the excited state atoms of neutral or inert gases (such as nitrogen or helium) generated by discharge resolve and instantly ionize the iconic compounds or compounds to be tested on the surface of the sample to be tested for mass spectrometry or tandem mass spectrometry detection to achieve real-time direct analysis of the sample [148]. Direct analysis in real-time mass spectrometry (DART-MS) is a typical direct mass spectrometry technique that can be used to obtain a spectrum within seconds from solid or liquid samples, and it has the advantages of stable performance and simple operation [155]. Since the DART-MS was reported in 2005, it has become one of the most widely used direct analysis methods. It is used to directly analyze the quality, safety, origin, and characteristics of food samples. Wang et al. [156] used DART-MS to analyze the flavonoid compounds in *Radix scutellaria*, optimized the gas temperature by using the peaks of the main active ingredients and wogonin, and discussed the ionization mechanism in detail. The results show that DART-MS can easily detect flavonoids due to the high proton affinity of compounds containing nitrogen or carbonyl. Huang et al. [157] established a combination of DART-MS, gas analysis–mass spectrometry (EGA-MS), and heart-cut EGA-gas chromatography (GC)/MS to characterize the chemical composition of Chinese crude propolis and showed that some flavonol glycosides, phenolic glycerides, and beeswax were easily pyrolyzed at 300–550 °C. Flavonoids with a larger molecular weight, such as quercetin (*m*/*z* 301) and rhamnose (*m*/*z* 315), cannot be detected using EGA-MS and heart-cut EGA GC/MS because of their strong polarity and high boiling points. Rýdlová et al. [158] combined the DART technique with TOF-MS. Determining and quantifying compounds characteristic of different cocoa products (theobromine, caffeine, phenols, and flavonoid compounds) were achieved by optimizing the ionization mode, ionization temperature, and solvent type of sample extraction. Combined with PCA analysis, it offers great potential for assessing the quality and authenticity of cocoa products. Lin et al. [159] developed untargeted and high-throughput methods for the rapid authentication of wine using MALDI-MS and DART-MS combined with OPLS-DA. Due to the interference of matrix ions, MALDI-MS is limited to the analysis of smaller compounds and is mainly used to detect anthocyanins and their metabolites. DART-MS mainly detects volatile compounds with small molecular weights. The complementarity of these two techniques allows for detecting more complex ions, demonstrating the feasibility of combining multiple mass spectrometry techniques for wine analysis.

### 3.4. Immunoassay Techniques

#### 3.4.1. Enzyme-Linked Immunosorbent Assay

Enzyme-linked immunosorbent assay (ELISA) is one of the most widely used analytical methods to quantify and isolate target compounds from complex substrates. Compared with instrumental methods, ELISA has a wider range of applications. In addition to being fast, simple, and effective, it requires less sample preparation and can test multiple samples simultaneously. The most common types of ELISA direct ELISA, indirect ELISA, sandwich ELISA, and competitive ELISA. Sandwich ELISA has higher specificity than the corresponding competitive analysis. However, sandwich ELISA requires two different antibodies to bind simultaneously to two antigen-binding sites on the desired analyte, so it is difficult to implement on molecules with molecular weights less than 1000 Da [160].

Qu et al. [161] developed a sandwich ELISA for detecting Naringin using two anti-Nar mAbs. These two hybridomas secreting anti-NAR monoclonal antibodies (mAbs) were generated by combining mouse spleen cells immunized with NAR-bovine serum albumin (BSA) with a mouse myeloma cell line sensitive to hypoxine-aminopterin-thymidine (HAT). The results showed that the LOD and LOQ of Nar dose–response curves were 6.78 ng mL ^−1^ and 13.47 ng ml ^−1^, and there was a high correlation between sandwich ELISA and HPLC (R^2^ = 0.9806). Sakamoto et al. [162] prepared the monoclonal antibody (MAb) against (DZ) and successfully applied it to develop an indirect competitive enzyme-linked immunosorbent assay (icELISA) for the simultaneous determination of DZ and genistin (GEN), which are known as two major soy isoflavone glycosides in soy products. Zhao et al. [163] synthesized a new naringenin hapten and developed an icELISA based on a naringenin monoclonal antibody for detecting naringenin content in pummelo and traditional Chinese herbs. The results showed that the linear range of detection was between 1.15 and 15.81 ng/mL. The findings of the icELISA for the analysis of naringenin correlated well with those of UPLC, showing good accuracy and reproducibility.

#### 3.4.2. Fluorescence-Linked Immunosorbent Assay

Fluorescence-linked immunosorbent assay (FLISA) is an immunoassay method based on fluorescein-labeled antibodies, which has the advantages of a low cost and high specificity and sensitivity. Compared with traditional ELISA, FLISA can avoid the time-consuming enzyme–substrate reaction required by ELISA and reduce the exposure time of the reaction system to the reaction temperature, greatly reducing the error [164].

Sakamoto et al. [165] developed a novel open-sandwich fluorescence-linked immunosorbent assay (os-FLISA) by taking advantage of enhanced interactions between variable regions of heavy- (VH) and light-chain (VL) domains in the presence of an antigen for the simultaneous determination of daidzin (DZ) and genistin (GEN) in soy products. Several results showed that os-FLISA is accurate and sufficiently sensitive, with a detection limit of 73.0 ng mL^−1^ for DZ and GEN. Shan et al. [166] labeled fluorescein isothiocyanate (FITC) in baicalin monoclonal antibody (MAb) and developed an indirect competitive fluorescence-linked immunosorbent assay (icFLISA). The LOD is 6.4 ng/mL to 500 μg/mL, which is two-fold lower than the LOD of the previously developed ELISA. Compared with the usual fluorescence immunoassay, time-resolved fluoroimmunoassay (TRFIA) can effectively avoid the interference of background fluorescence and greatly improve the sensitivity and accuracy of detection, and it has been widely used in the medical field. Uehara et al. [167] developed a TRFIA for the quantitative determination of enterolactone, genistein, and daidzein in human urine. The results showed a significant correlation between immunoassay and GC ± MS results. The method is rapid and accurate and has been applied to the analysis of phytoestrogens in human urine.

#### 3.4.3. Immunochromatographic Assay

The immunochromatographic assay is mainly completed with the help of immunochromatographic test strips. It is simple to operate, has rapid detection and strong specificity, and is convenient to perform. Common immunochromatographic test strips include colloidal gold test strips and quantum dot test strips, which can greatly simplify the detection procedure and avoid prolonged incubation and complex operation steps.

Paudel et al. [168] prepared a colloidal gold-conjugated anti-baicalin monoclonal antibody (anti-BA MAb) and developed a method for the rapid one-step detection of baicalin by immunochromatographic assay combined with ic-ELISA. The detection limit is 200 ng/mL ~2 μg/mL, and the detection time is only 15 min. This method is simple, cheap, and easy to use, and the effectiveness of baicalin analysis in *Scutellariae radix* and Kampo medicines has been proven. Sakamoto et al. [169] established a one-step ICA using gold nanoparticles conjugated with a monoclonal antibody for the rapid and sensitive detection of total isoflavone glycosides. ICA completes this determination within 15 min of dipping the test strip into the analyte solution, with an LOD of 125 ng mL^−1^. Qu et al. [170] established a rapid and quantitative lateral-flow immunoassay using a quantum dots (QDs)–antibody probe to analyze puerarin. The results showed that the method could determine puerarin content in water and biological samples within 10 min. The LOD of the method was 1–10 μg/mL, the IC_50_ value was 75.58 ng/mL, the LOQ was 5.8 ng/mL, and the recoveries were 97.38–116.56%.

### 3.5. Overview of Research Progress on Flavonoids Detection

At present, the detection of flavonoids mainly focuses on instrumental analysis. Spectroscopic and chromatographic techniques are the most commonly used methods for qualitative and quantitative analysis of flavonoids, and each has its advantages. For example, UV–Vis spectrophotometry is simple to operate and fast to analyze and is suitable for the rough determination of the total flavonoid content in foods and plants. However, it can only detect the information for specific groups in the molecule and cannot obtain the information for all compounds. Fluorescence analysis has higher sensitivity than UV–Vis, but the number of flavonoids with natural fluorescence is limited, and additional sample-handling procedures are required to form fluorescent complexes. NIR spectroscopy is a simple, fast, accurate, and non-destructive technique that can be combined with HIS to obtain more comprehensive and effective information. However, NIR spectroscopy often needs to be combined with chemometrics to build models, and the number of data is large, and processing them is time-consuming. HPLC has the advantages of a wide detection range, low detection limit, and high sensitivity, and it is usually combined with DAD, UV, FS, or MS. For example, the liquid chromatography–mass spectrometry technology integrates the high separation ability of liquid chromatography and the structural analysis ability of mass spectrometry, which has the characteristics of high sensitivity, high resolution, fast speed, and stability, and it is an important technical means for the simultaneous qualitative and quantitative analysis of complex compounds such as flavonoids. Because multi-step sample pretreatment must be performed before sample analysis and the separation of multiple components requires a long chromatographic run time, it is time-consuming and has difficulty meeting the requirements of rapid analysis. The particle size used by UPLC is less than 2μm, and due to the small particle size, the diffusion path between the fixed phase and the analyte is shorter. The analysis speed, resolution, and sensitivity are significantly improved. SFC uses supercritical fluid as the mobile phase, and the critical fluid has the characteristics of high solubility, high diffusivity, and low viscosity, which can achieve fast and effective separation. Chromatographic methods are sensitive and specific, but there are some obvious shortcomings, such as complex pretreatment, long determination time, and high cost, which limit their wide application in rapid screening detection to a large extent. As a new technology, direct mass spectrometry can obtain the information of compounds within a few seconds without sample pretreatment. It has the advantages of high stability and selectivity and can realize the rapid analysis of trace flavonoids in foods.

Instrumental analysis is limited by requiring expensive instruments and skilled inspectors. In contrast, immunoassay has the advantages of simple operation, high sensitivity, and a low cost of analysis, which makes it a very promising detection method. In recent years, researchers at home and abroad have performed a lot of exploration and research in the preparation of flavonoids haptens and antibodies and have made certain progress. The antibodies of flavonoids have been successfully applied in the establishment of various immunoassay methods, such as the enzyme-linked immunoassay method and the fluorescence immunoassay method, which can realize the qualitative and quantitative detection of specific flavonoids. However, there are still many problems in the immunoassay of flavonoids, and most flavonoids have not produced polyclonal or monoclonal antibodies. Although some immunoassay methods have been established, there are still some problems in practical applications, such as low sensitivity, poor specificity, and large matrix interference. Therefore, according to the structural differences of flavonoids, the preparation of antibodies with high sensitivity and specificity and the improvement of the detection accuracy of actual samples will be the future direction of immunoassay methods. In addition, with the continuous improvement in the original technology and the combination of stoichiometric methods, more and more determination methods have been gradually established, and high precision, short time consumption, simple operation, low cost, and low pollution are the inevitable trend of the development of determination methods.

## 4. Conclusions

Flavonoids have a variety of applications in the food industry, such as preservatives, colors, and antioxidants, as well as in other industries such as pharmaceuticals. Research in all these areas relies heavily on accurate analytical data, so it is important to develop high-purity and efficient extraction methods and effective and rapid detection methods. Traditional extraction techniques are still used because they do not need special equipment. However, further purification is often required due to its longer extraction time, low purity, and low efficiency. Modern extraction methods, including UAE, MAE, EAE, SFE, and ASE, have been widely used to replace traditional methods because of their advantages, such as high extraction rate, high selectivity, stability of the target extract, and processing safety. The loss of the target product occurs at each step of the extraction process of flavonoid compounds. Therefore, it is necessary to consider the properties of the target compound and optimize the extraction conditions when selecting the extraction process to improve the recovery rate as much as possible.

Flavonoids have been detected and analyzed using spectroscopy, chromatography, mass spectrometry, and immunoassay. Each of these methods has its advantages, among which is that direct mass spectrometry has great potential as a new technique that can obtain information about compounds in a few seconds without sample pretreatment. It also has the advantages of high stability and selectivity and can realize the rapid analysis of trace flavonoids in complex substrates. Immunoassay does not require expensive instruments or professional detection personnel. It has the advantages of simple operation, high sensitivity, and low analysis cost, which make it a very promising detection method. However, most flavonoids have not yet produced polyclonal or monoclonal antibodies, which makes immunoassay limited in its practical application.

Chromatographic methods usually require cleaning the sample before analysis, and such an operation may cause a loss of the target product. The traditional pretreatment method for chromatographic analysis is usually organic solvent extraction, which consumes a large amount of organic solvent and interferes with the sample peak. Combining modern extraction techniques and chromatographic analysis can reduce the loss of target compounds and thus increase the recovery rate. In summary, with new extraction techniques and analytical methods, more and more flavonoids in foods or plants have been isolated, identified, and studied. Nevertheless, continuous efforts are needed to develop more analytical techniques with potential for development to improve the bioavailability of flavonoid compounds in food and promote the application of flavonoid compounds in the food and pharmaceutical industries.

## Figures and Tables

**Figure 1 foods-13-00628-f001:**
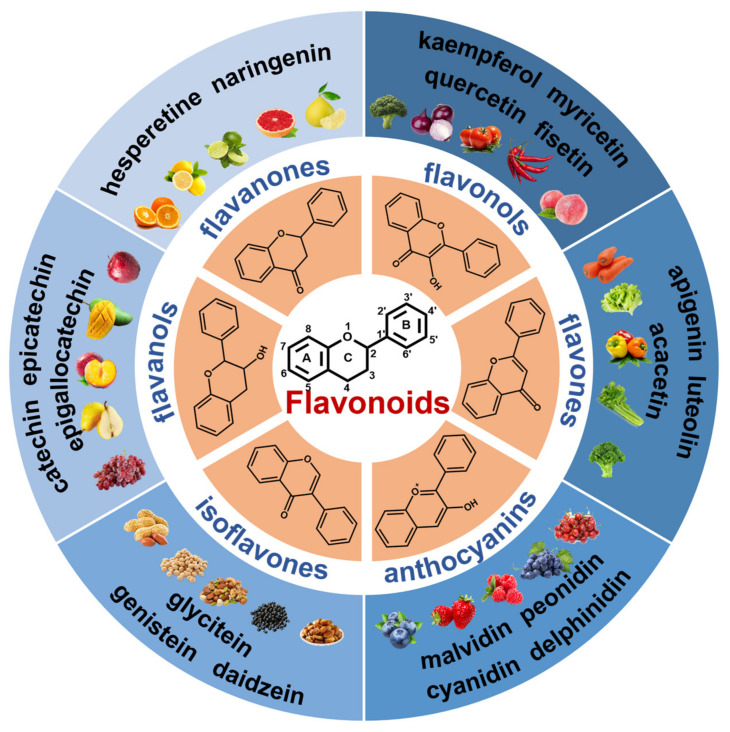
Basic chemical structure of flavonoids, different classes, and their food sources.

**Table 1 foods-13-00628-t001:** Emerging techniques used in recent years for the extraction of flavonoids.

Extraction Method	Substrate	Optimal Extraction Conditions					Yield	Extraction Efficiency	Ref
		Solvent	Time	Solid-to-Solvent Ratio	Temperature	Others			
MAE	Parsley leaves	ethanol 80%	2 min	1:20	-	particle size: 0.105 mm microwave power: 180 W	Apigenin (7.90 ± 0.14 mg/g)	42.68%	[26]
MAE	*Sedum aizoon* leaves	ethanol 80.6%	20 min	1:20,	57 °C	-	Flavonoids (20.01 ± 0.3 mg/g)	-	[27]
MAE	*Syzygium nervosum* Fruits	ethanol	38 min	1: 35	-	microwave power: 350 W	2,4-Dihydroxy-6-metoxy-3,5-dimethyl chalcone (1409 ± 24 μg/g)	-	[28]
MAE	*Melastoma sanguineum* Fruit	ethanol 31.33%	45 min	1: 32.21	-	microwave power: 500 W	TPC (39.02 ± 0.73 mg GAE/g DW)	-	[29]
MAE	Soybean meal	ethanol 50%	0.16 min	1:60	-	microwave power: 120W	TPC (13.09 mg GAE/g) TFC (7.39 mg CE/g)	-	[30]
MAE	Lemon myrtle	aqueous acetone 50%	6 min	6:100	-	microwave power: 630 W	Flavonoids (384.57 ± 2.74 mg CE/g DW) Proanthocyanidins (336.54 ± 7.09 mg CE/g DW)	-	[31]
MAE	Grape skin	ethanol 60%	5 min	-	40 °C	microwave power: 600 W	Total anthocyanins (12,545.19 mg/kg)	-	[32]
DES-MAE	*Flos sophorae immaturus*	DES: choline chloride/1,4-butanediol (molar ratio 1:2), water content: 25%	20 min	1:26	62 °C	microwave power: 600 W	Rutin (116.78 mg/g) Nicotiflorin (15.01 mg/g) Narcissin (23.85 mg/g) Quercetin (27.59 mg/g) Kaempferol (3.09 mg/g) Isorhamnetin (3.33 mg/g)	-	[33]
DES-MAE	*Ribes* *mandshuricum* leaves	DES: choline chloride/lactic acid (mass ratio 1:2), water content: 25%,	10 min	1:27	54 °C	-	Trifolin (4.78 mg/g DW) Isoquercetin (2.57 mg/g DW) Rutin (1.25 mg/g DW) Astragalin (1.15 mg/g DW) Quercetin (0.34 mg/g DW) Hyperoside (0.32 mg/g DW) Kaempferol (0.093 mg/g DW)	-	[34]
NADES-MAE	Sweet potato leaves	DES: Choline chloride/Malic acid 1:1	21 min	1:70	54 °C	microwave power: 470 W	TFC (40.21 ± 0.23 mg RE/g)	-	[35]
UAE	Purple sweet potatoes	ethanol 90%	60 min	-	60 °C	ultrasonic power: 200 W	Anthocyanins (214.92 ± 11.59 mg/100 g DW)	-	[36]
UAE	Date palm spikelets	ethanol 70%	21.6 min	-	40.8 °C	ultrasonic power: 110 W frequency: 40 kHz	Rutin (114.6 ± 2.29 mg/g) Quercetin (12.0 ± 0.58 mg/g) Kaempferol (0.3 ± 0.01 mg/g)	-	[37]
UAE	Kiwifruit	ethanol 68%	30 min	1:20	40 °C	ultrasonic power: 420 W	TFC (5.10 ± 0.09 mg CE/g DW)	-	[38]
UAE	Mushrooms	MeOH 93.6%	5 min	-	60 °C	amplitude 16.86%, cycles 0.71 s^−1^.	-	-	[39]
UAE	Peanut shells	ethanol 70%	-	1:40	55 °C	particle size: 0.285 mm, ultrasonic power: 120 W, frequency: 45 kHz	9.26 mg/g	-	[40]
UAE	Parsley leaves	ethanol 80%	30 min	1:25	40 °C	particle size: 0.25 mm ultrasonic power: 90%, frequency: 80 kHz	Apigenin (9.48 ± 0.11)	51.22%	[26]
UAE	Apples	ethanol 70%	26.90 min	-	44.61 °C	ultrasonic power: 480 W	Rutin (6.58 mg/g)	-	[41]
UAE	Watermelon rind	acetone 70.71%	10.65 min	1:30.50	29.78 °C	ultrasonic power: 300 W	TPC (6.31 ± 0.14 mg GAE/g db) TFC (3.16 ± 0.08 mg Rutin/g db)	-	[42]
UAE	Green coconut shells	-	15 min	1:24	33 °C	-	TPC (40.99 GAE/g) TFC (36.13 QE/g) Total Tannin content (176.73 TAE/g)	-	[43]
DES-UAE	Grape skin	DES: choline chloride-based DES containing oxalic acid with 25% of water	50 min		65 °C	-	-	-	[44]
DES-UAE	*Lycium barbarum* L. fruits	DES: 1:2 mixture of choline chloride and p-toluene sulfonic acid	90 min	1:20	25 °C	-	Myricetin (57.2 mg/g) Morin (12.7 mg/g) Rutin (9.1 mg/g)	-	[45]
NADES-UAE	Mangosteen rind	NADES: 1:2 mixture of 1,2-Propanediol and Lactic acid with 30.3% of water	9.1 min	1:76.7	57.5 °C	-	TPC (118.2 ± 3.8 mg GAE/g db) TFC (59.3 ± 2.1 mg RE/g db)	-	[46]
EAE	*Capparis* *spnosa* fruit	commercial enzyme mixture composed of β-glucanase, xylanase, cellulase, α-amylase, and protease, enzyme concentration: 6.5%	60 min	-	50 °C	pH: 7.5	TPC (24.76 GAE/g) TFC (24.56 mg CE/g)	-	[47]
EAE	Corn tassel	1:1 mixture of cellulase and protease	-		50 °C	pH: 5.0	TPC (10.70 mg/g)	-	[48]
EAE	Blackcurrant press cake	β-glucanase	-	1:10	50 °C	pH: 5.5	TPC (−1142 mg/100 g)	-	[49]
EAUMSE	Chinese water chestnut peels	2:1 mixture of cellulase and pectinase, enzyme concentration: 1.5%, hydrolysis temperature: 50 °C, hydrolysis time: 2 h, extraction solvent: ethanol and 0.1 mol/L NaH_2_PO_4_ buffer (2:1, *v*/*v*) mixture	60 s	-	-	pH: 5.0 ultrasound power: 50 W, microwave power: 200 W	Luteolin (249 mg/100 g DW) Eriodictyol (97 mg/100 g DW) 6-methylluteolin (74 mg/100 g DW) Fisetin (46 mg/100 g DW)	-	[50]
E-UAE	Pomelo peel	enzyme concentration: 2%	60 min	1:40	50 °C	ultrasonic energy: 40 kHz	TPC (1.76 ± 0.06 mg RE/g) (1.15 ± 0.04 mg RE/g) (2.29 ± 0.05 mg RE/g)	-	[51]
ASE	Purple sweet potatoes	80% (*v*/*v*) aqueous ethanol containing 0.1% (*v*/*v*) HCl	15 min		90 °C	the number of cycles: 2	Anthocyanins (252.34 ± 10.59 mg/100 g DW)	-	[36]
ASE	Goji berry	ethanol 86%	20 min		180 °C	pressure: 10 MPa (1500 psi)	TF (3.02 mg/g)	-	[52]
ASE	Canola meal	ethanol 70%	-	-	180 °C	-	TPC (24.71 ± 2.77 mg SAE/g DM)	-	[53]
ASE	Strawberry tree fruit	ethanol 96%	10 min	-	120 °C	the number of cycles: 2	-	-	[54]
ASE	Mung bean seed coat	ethanol 50%	-	-	-	pressure: 1300 psi	TPC (55.27 ± 1.14 mg GAE/g) TFC (934.04 ± 0.72 mg CE/g)	-	[55]
SFE-CO_2_	Xinjiang jujube	-	113.42 min	-	52.52 °C	pressure: 27.12 MPa cosolvent flow rate: 0.44 mL/min	29.05 ± 0.38 mg/g	-	[56]
SFE-CO_2_	Tomato skin	-	80 min	-	60 °C	CO_2_ flow rate: 2 mL/min pressure: 550 bar	Lycopen (0.86 ± 0.06 mg/100 g DW) β-Caroten (1.5 ± 0.4 mg/100 g DW)	-	[57]
SFE-CO_2_	Mango seed kernels	ethanol 15.0%	-	-	60 °C	pressure: 21.0 MPa	TPC (19.4 mg-eq AG g^−1^ extract), TFC (3.8 mg-eq Q g^−1^)	-	[58]
SFE-CO_2_	Bentong ginger	-	-	-	40 °C	pressure: 25.0 MPa particle size: 300 µm	6-gingerol content (171.26 mg/g) TPC (17.84 GAE mg/g) TFC (74.46 QE mg/g)	-	[59]
SFE-CO_2_	Rice husk	25% ethanol-water (50%, *v*/*v*) cosolvent mixture	-	-	60 °C	pressure: 30 MPa	TPC (1.29 mg GAE/g) TFC (0.40 mg CE/g)	-	[60]

Abbreviations: GAE: gallic acid equivalent; DW: dry weight; CE: catechin equivalent; RE: rutin equivalent; SAE: sinapic acid equivalent; QE: quercetin; MAE: microwave-assisted extraction; UAE: ultrasound-assisted extraction; DES-MAE: deep eutectic solvents-microwave-assisted extraction; NADES-MAE: natural deep eutectic solvents–microwave-assisted extraction; DES-UAE: deep eutectic solvents–ultrasound-assisted extraction; NADES-UAE: natural deep eutectic solvents–ultrasound-assisted extraction; EAE: enzymatic assisted extraction; EAUMSE: enzyme-assisted ultrasonic-microwave synergistic extraction; E-UAE: enzyme and ultrasound-assisted extraction; ASE: accelerated solvent extraction; SFE-CO_2_: supercritical fluids extraction–CO_2_; TPC: total phenolic content; TFC: total flavonoid content.

**Table 2 foods-13-00628-t002:** The advantages and disadvantages of the various analytical techniques.

Detection Techniques		Advantages	Disadvantages	Flavonoid Compounds
Spectrometric techniques	UV–Vis	Rapid analysis, low cost, simple operation	The interference of other components in the complex matrix will affect the accuracy of detection, and only the information of specific groups in the molecule can be detected, but the information of all compounds cannot be obtained	Quercetin Rutin Luteolin
	FS	High sensitivity, good selectivity, simple operation, and fast detection	The number of naturally fluorescing flavonoids is limited, fluorescence quenching effect, and scattering light interference problems	Catechin Anthocyanins Quercetin kaempferol
	NMR	Fast, non-destructive, high stability and reproducibility, simple sample preparation, can provide rich structural information, qualitative and quantitative ability is excellent	Lack of sensitivity, high cost, specific compounds cannot be isolated, complex samples may have signal overlap problems	Naringenin Hyperoside Rutin Diosmetin Luteolin Calycosin
	NIR	Fast, non-destructive, no need to pre-treat the sample, no damage to the tested sample, real-time online detection can be achieved, combined with hyperspectral imaging technology, spectral and spatial information of samples can be obtained at the same time, and the accuracy of detection can be further improved	Need to be combined with chemometrics to establish the model of the sample data quantity is large, high-cost modeling	Anthocyanin Chrysin Galangin Hyperoside Isoquercitrin
Chromatographic techniques	SFC	High separation efficiency, short extraction time, less consumption of organic solvents, and controllable extraction conditions	Require expensive special material container to maintain high-voltage operating conditions	Apigenin Baicalin Luteolin Naringenin
	HPLC UPLC	Low detection limit, high sensitivity, high analysis efficiency, and wide analysis range	High detection cost, long detection time, complex operation, and high solvent consumption	Quercetin Kaempferol Isorhamnetin Rutin Naringin Hesperidin Nobiletin Tangeretin
Mass spectrometry	LC-MS	Efficient separation performance and high sensitivity	High detection cost and expensive equipment	Luteoloside Rutin Hyperoside Quercetin
	EESI-MS DART-MS	No sample pretreatment is required, and analyte information can be obtained in seconds. Without complex sample metabolite separation processes, more comprehensive metabolic information can be obtained	Need to consider the interference of matrix ions in the sample, equipment is expensive, difficult to achieve large-scale industrial application	Naringin Apigenin Quercetin Myricetin Kaempferol Rutin Epicatechin Hesperetin
Immunoassay technique	ELISA FLISA TRFIA ICA	No need for expensive equipment or professional operators, simple operation, high sensitivity, and low analysis cost	High specific antibody preparation is difficult, the specificity of the assay, only one compound can be detected	Daidzin Genistein Glycitein Naringenin Hesperetin Isoxanthohumol Apigenin-C-glycoside Luteolin glycosides

Abbreviations: UV–Vis: ultraviolet–visible spectrophotometry; FS: fluorescence spectroscopy; NMR: nuclear magnetic resonance; NIR: near-infrared Spectroscopy; HPLC: high-performance liquid chromatography; LC-MS: liquid chromatography–mass spectrometry; UPLC: ultra-performance liquid chromatography; SFC: supercritical fluid chromatography; EESI-MS: extractive electrospray ionization–mass spectrometry; DART-MS: direct analysis in real-time mass spectrometry; ELISA: enzyme-linked immunosorbent assay; FLISA: fluorescence-linked immunosorbent assay; TRFIA: time-resolved fluoroimmunoassay; ICA: immunochromatographic assay.

## Data Availability

No new data were created or analyzed in this study. Data sharing is not applicable to this article.

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
