# Peer review of "Research Progress on Extraction and Detection Technologies of Flavonoid Compounds in Foods"

_foods, 2024, doi:10.3390/foods13040628_

Round 1

Reviewer 1 Report

Comments and Suggestions for Authors

The present manuscript is a review on the progress on extraction and detection technologies for flavonoids in foods. It is a relevant topic and the authors cover many aspects. However, in my opinion the main drawback of the review is that, as its current form, is does not fulfill its objectives, i.e., it does not really provide the reader a clear view on current progresses in the field. In each heading, the authors comment some specific articles, which are recent studies, but sometimes they are using techniques that have been applied in the field of flavonoids for years and, sometimes they are new, but it is really hard to know the case for each one. Also, since the review aims to be focused on research progress, there is no need to comment general aspects for each of the methodologies which are mentioned, but the review is always at an intermedium position, where it is not clear whether it aims to be a general guide for analyzing flavonoid or a review on research progress. Besides, there is no perspectives nor specific output of the review, providing an overview of current tendencies, limitations, etc., but it is more a long list of specific studies. For these reasons, I would not recommend the paper for publication. Nevertheless, I provide a detailed list of modifications that could be performed in order to prepare a new submission:

General comments

- Heading 2. At each sub-heading, the authors should highlight the novel aspects of the studies they select, in contrast to the most common ones. As I stated above, since the review is focused on research progress, it must be assumed that the reader already knows at least the general aspects of each technique, so there is no need to describe each one of them, unless some specific technical aspect is related to one of the selected publications (and it would be discussed when talking about that publication, and not as a short introduction to each technique). Also, at the end of the heading the authors should include an additional sub-heading called “Overview of research progress on flavonoid extraction”, where they provided (and discussed) a figure with a workflow for flavonoid extraction based on current knowledge. This should provide useful information to the reader and it would be a clear output of the review. Otherwise, the reader will be overwhelmed by the abundance of specific studies but lacking some guidelines. Also, in this sub-heading the authors should briefly mention the aspects that remain to be explored for improving flavonoid extraction. And they should also mention that there is a fraction of non-extractable polyphenols, including flavonoids, which are known not to be obtained by any extraction procedure, remaining in the corresponding residues.

- Heading 3. Similarly to heading 2: a) there is no need to provide a short introduction to each technique; 2) the authors should make clear, for the selected publications, why they represent a progress in the field; 3) there should be a last sub-heading integrating all the information in the way of practical guidelines for the reader.

- Table 2. Although I consider such a table is not needed in a review focused on current progress (instead of general considerations for analysis), I disagree with many aspects in it: 1) there are several methods for flavonoid analysis based on UV-vis, but they are not characterized by a high sensitivity nor selectivity; 2) the main disadvantage of NMR is associated with the problems for isolating specific compounds, particularly in the context of flavonoids, where a sample may contain dozens of different compounds; 3) I disagree that a characteristic of HPLC is that sample processing is demanding (commonly, it is similar to that for UV-Vis analysis with additional steps of concentration and filtration, which are not a huge modification); 4) HPLC and UPLC should share advantages/disadvantages, showing as differences that UPLC requires shorter times and less solvent volume; 5) I am surprised to see TLC, which is not use anymore as an analytical technique per se, but in combination with other systems, as the authors discuss below; regarding MS, it is not right that it does not allow to detect isomers and its main advantage, i.e., the ability to identify compounds without standards, is not mentioned. Besides, the reference column should be deleted: it is not needed for such a table and it is based on a mixture of general papers on a technique and specific studies (only general papers should remain as references, as foot note). And there is a lack of consistency on the style used for most techniques, based on short statements, and the ones used in MS (LC-MS and EESI-MS, DART-MS) with long sentences.

- English should be reviewed throughout the manuscript. In particular, there is a tendency to wrongly use singular when it should be plural (e.g., “it is commonly found” in line 35 when it should be “they are commonly found”; “it is commonly” in line 44 when it should be “they are commonly”).

- The authors should revise the consistency in the use of abbreviations. Currently, some abbreviations are provided twice (such as ANN), the same abbreviation may be used for different concepts (such as TFC in lines 86 and 93) or there are concepts for which abbreviations have already been provided but which are later written in full (total flavonoids, mass spectrometry, etc.)

Specific comments

- Figure 1 appears twice.

- Lines 39-42. This should be deleted: since this paragraph provides a general overview of flavonoids, it has no sense to focus on the biological activities of a particular polyphenol.

- Lines 54-57. Idem.

- Line 47. When describing flavanols, the authors should explain the difference between monomeric flavanols (catechins) and oligomeric/polymeric flavanols (proanthocyanidins).

- Line 60. “indispensable” is a too strong word.

-Lines 72-78. Since this review is focused on isolation7analytical aspects, I think these particular examples may be deleted and it is enough with the previous general statement.

- Line 100. “Regarding analysis” should be added before “Spectrometric and chromatography techniques”.

- Lines 107-08. Based on current writing, it seems that HPLC may be coupled either with DAD or with MS, which is not right: as detector, DAD is the most common one (but not the only one, as they mention later): And then, additionally, it may be combined with MS. Sentence should be rewritten to avoid this misunderstanding.

- Table 1 should be discussed and not just mentioned. The authors should transmit the most relevant facts from it since, currently, it is only a list of publications. Besides, it should be moved to the beginning of heading 2.1 since it corresponds to that heading.

- Line 240. “Overall,” should be added before “Compared”.

- The Conclusions section is too long (nearly one page). The authors should summarize the main messages arising from their review in one paragraph.

- I think the authors should deserve some space to the analysis of monomeric/oligomeric flavanols, due to its specific characteristics.

Comments on the Quality of English Language

English should be reviewed throughout the manuscript. In particular, there is a tendency to wrongly use singular when it should be plural (e.g., “it is commonly found” in line 35 when it should be “they are commonly found”; “it is commonly” in line 44 when it should be “they are commonly”).

Reviewer 2 Report

Comments and Suggestions for Authors

In general, the manuscript is interesting and provides enough information in a single article. It is important that you take into account the following recommendations:

Figure 1 appears two times

Table 1 is very complete and is very helpful in reaching the reference to isolate a group of specific flavonoids. You should check the names of the compounds carefully because there are some with a capital letter and others with a lower case letter, fix that.

The conclusions are very long, perhaps it would be good to summarize a little

Reviewer 3 Report

Comments and Suggestions for Authors

see pdf file

Comments on the Quality of English Language

 Minor editing of English language required

Reviewer 4 Report

Comments and Suggestions for Authors

The review "Research Progress on Extraction and Detection Technologies of Flavonoid Compounds in Foods" is well-written and covers the topic comprehensively. The summarization of the extraction, detection, and spectroscopic technique in a tabular format is presented well. A few concerns are underlined below.

1.  Pyran ring is 6 membered ring with C5 and not C3 (Line27)

2. Figure 1 is repeated twice.

Minor: 

Thorough revision for typos is required e.g. (7.216mg/g) should be (7.22 mg/g) 

Edit Table 1 for minor typos

Compared with s single-enzyme- assisted extraction, more phenolic compounds were quantitatively? What is s???

Comments on the Quality of English Language

Moderate editing is required

Reviewer 5 Report

Comments and Suggestions for Authors

The Manuscript is entitled: Research Progress on Extraction and Detection Technologies of Flavonoid Compounds in Foods and is proposed by: Li W, Zhang X, Wang S, Gao X and Zhang X.

The aim of the work is to present and compare various extraction and detection techniques of flavonoids from plant matrix. The main problem is that the comparison between methods is difficult since yield are not expressed as percentages but as raw concentrations. Since many genetic and environmental factors can influence the flavonoid content of a plant, the review is not informative enough. Therefore the authors should indicate percentages of yield and detection comparing conventional and optimized conditions. This could better help the future readers.

In addition, the authors did not considered the immunological methods that were currently developed to assay flavonoids in plants. Their sensitivity equal that of LC MS/MS techniques and allow to assay active flavonoids in many matrix including plants, transformed food but also consumers tissues like urine, serum or even hairs. The author should examine this kind of work before re-submitting their review.

Line 25. I do not agree with the statement saying that glycosylation increase the transport across the cell membrane. Indeed, glycosylation increase the flavonoids water-solubility and therefore decreases their affinity for the lipidic cell membranes. However, specific active transporters exist at the cell membranes that are involved in flavonoids cell membranes crossing. Therefore, it is required to specify that the transport of glycosylated flavonoids is an active transport via specific enzymes that could be named. To sustain this statement the authors can rely on the bioavailability of flavonoids in human body fluids.

It seems that figure 1 appears twice and the first one should be conserved with the caption which is present in page 3. Please detail in the caption the sources of isoflavones that have been represented. Indeed, it is very difficult to identify them from the pictures provided. Peanuts, lentils and maybe soybeans could be identify but the two other sources are unclear.

Presently, it is unclear if the paragraph from line 33 to 57 is a caption of figure 1. If it is please remove the health effects that are described not only some of the compounds because it is unclear why some compounds are evocated and not others.

From line 60. Please when scientific publications are cited to illustrate a health effect it is crucial to mention if the data were obtained in vitro or in vivo. Indeed, the bioavailability of flavonoids is highly variable according to the substance considered from 4% to 80%. Such variability has a tremendous effect on health properties but cannot be estimated in invitro studies especially when they are performed with compounds far from the biological forms and concentrations.

Antioxidant effects should be demonstrated in vivo and not as a oxidative measurement but rather as the results of the prevention of oxidation of inner molecules. A low level of oxidized blood lipid can be a good biomarker.

Line 86-87. “play a key role in flavonoid 86 utilizing for human heathy” should be replaced by “play a key role in the action of flavonoids used for human health

Paragaph 2.1. Conventional extraction

Please mention that the LogP values of biological substances can help choosing the right solvent providing that the current form in plants is taken into account. I mean that if the substance is mainly under a glycosidic form with a low LogP, the extraction will be essentially efficient with polar solvent and usually with water.

A liquid-liquid extraction with two non-miscible solvents with two different polarities (ethyl-acetate / water for instance) can be very efficient in flavonoids, isoflavonoids of lignans extraction. It does not require large solvent volumes and can be automatized and applied on large series of samples including plant, food, human biological fluids or hair samples.

Line 156: “Matricaria chamomilla” should be written in italic case.

Line 164: “Pleioblastus amarus” should be written in italic case.

Line 170: “Artocarpus heterophyllus” should be written in italic case.

Table 1. Plant latin names should always be written in italic cases and the gender name should be written with a capital case.

Table 1. The Yield column seems un-fare. A yield should be express as a percentage of spike concentrated compound. It is known that this method is not perfect but at least it can be comparable. Therefore please precise the % and the compound used as reference. Can the author change this column or add such data in a new column?

Line 226: “V. amygdalina” should be written in italic case and the gender Vernonia should be specified since it was never cited before.

Line 241: remove “under”

Line 303: I guess that the authors mean: “compared to a single enzyme-assisted-extraction”

Line 307: “Capparis spinosa” should be written in italic case.

Line 351-352: “Vigna radiata” should be written in italic case.

Line 355-356: “Arbutus unedo” should be written in italic case.

Line 384: “Mangifera indica” should be written in italic case.

Line 412: please correct “conditions,Choline” there is a space missing and choline should be written without capital letter.

Line 413 and elsewhere in the text. When extraction methods are compared it is better to give a percentage of efficacy rather than a raw value that does not bring any precise information. Indeed, the amount of flavonoids in plant depends on many ecological factors and therefore a raw value in mg/g for instance is not informative enough.

Line 415 and 419: “Lycium barbarum” should be written in italic case.

Line 429: “Ribes mandshuricum” should be written in italic case.

Line 430: percentage of extraction in reference to classical extraction should be better used.

Line 441: Please you should precise that chromatographic methods and especially HPLC cannot accept crude extracts. Usually the samples should be cleaned before analysis and such operation may remove some compounds of interest. Therefore, although HPLC is the technique most used for flavonoid analysis it may not be as exhaustive as other method which can be used directly on raw material.

Table 2. It seems that the authors forgot the immuno-assay. Some have been developed for flavonoids, isoflavonoids and lignans and allow detection and quantification in samples of different origin. In addition they can be very sensitive and highly specific. They are simple and cheap once set up and can be used automatically even by students. However, they are highly specific and does not allow the measurements of several compounds at once.

Line 457-472: I wonder if a mention to DAD detection that can ensure better identification of a flavonoid substance should not be done in this paragraph dealing with UV-vis detection…

Line 488-489: “Allium sativum” should be written in italic case

Line 493: “A. sativum” should be written in italic case.

Line 530: “Crataegus” should be written in italic case.

Line 537: “A. annua” should be written in italic case The gender Artemisia should also be mentioned since it has never been mentioned previously.

Line 615: I suggest to add a comment on the publication DOI: 10.3390/foods11020232.

Line 670: DAD has not be defined previously.

Line 671: please add a space in the adequate position “Citri Reticulatae Pericarpium(CPR).”

Line 675-676: Beware of the notion of good sensitivity. A sensitivity of a few µg/mL is far less than what can be obtained in immunoassay or in LC-MS-MS. You can indicate a figure but it seems to me that the word “good” is not relevant.

Line 686-691: The sentence is too long and difficult to understand. I suggest to modify it as follow: High performance liquid chromatography (HPLC) is a technique that separates the mixture in the mobile phase through the chromatographic column. It can be used with different detectors which can also characterize the sample. The latter is separated by the chromatographic column and enters the detector with the mobile phase. Then the detector converts the physical or chemical signal of the sample into an electrical signal to obtain a chromatogram of the different sample components.

Line 707-708: I guess the author meant Artemisia annua which should be written in italic case.

Line 718: “Abrus precatorius” should be written in italic case.

Line 720: please express the difference in percentage of the classical method.

Line 736-737. It should be mentioned that samples need to cleaned before analysis and this cleaning steps can reduce some of the native compounds under the detection limit.

Line 742. Could the authors give some precisions on the Photo-diode array detection method?

Line 814-815: a space is missing in “Desorption Electrospray Ionization Mass Spectrome-814 try(DESI-MS)” and in “Extractive Electrospray Ionization Mass Spectrometry(EESI-MS)”

Line 852: “Anoectochilus roxburghii” should be written in italic case.

Line 870: Please check the name “Scutellariae Radix” that could well be Scutellariae baicalensis or Radix scutellaria. In all cases this Latin name should be written in italic case.

Line 902: it seems something is wrong in the sentence “get a lot of products and still be selected”

Conclusion: the authors should differentiate the extraction methods tending to recover as many substances as possible to preserve for instance the crude plants’ health properties and extraction methods required to clean samples before chromatographic separations. Indeed, sometimes to preserve analytical column a complex mixture should be simplified and this may have an impact on its final analytical knowledge.

Immuno-techniques should be mentioned in the conclusion with their advantages and limits.

Comments on the Quality of English Language

The Latin names should be written in italic case all over the manuscript. There are also a few typing mistakes.

Round 2

Reviewer 1 Report

Comments and Suggestions for Authors

I think the manuscript has clearly improved with the modifications included by the authrosw. However, may mian cristicism remains: I think this is more a review on the analysisi of flavooids (a topic on which tehre are many toher reviews avaiable) than on research progress (which is supposed to be itas aim). I ill try to make clearer this psoition:

- Technqiues asuch as MAE and UAE have lerady been used for some decades, so including them in areview does not make the review to be autoamttically focued in resrach progress.

- Selecteding recent publciaitons does tno make autoamttically that the review is fous on reserach progress 8form ana anlyticla point of view), unless the turhos celarly state the novel aspects from thatose studies. Otherwise, the aturhos may be using, for intance, a common HPLC-MS method for analysisn a fruti not previoysly characterized, but this does not consitute somehting novel within the frame of this review.

- Althought he authos included a heading at the end of each section providign an overview of the progresses, indeed they rpovided auite genral commetns that may ahve been written 15 eyrs ago and they do not really creflect advances in teh filed.

For these reasosn, I still consider the amnsucirpt would request a deep revision i order to amke it sutiabalbe for pbulcaiton.

Specific commetns on Table 1

- I am suprosed to see the last column with compoudsn analyzed by each emthod. This only makes ense int ehc se of inmunochemcial emthods where a few methods have beend eveloepd. But it shoudl not be icnldued for the other methodologies, since most of them are applciable to most falvooids. I would suggest to revmove this column.

- The atuhros include as a disdvnte of HPLC/UPLC that "The celanign step of the ample prior to analysis results in teh loss of the target compound". I fully disagree with this. It may happen in some specificc case, but it is not at all a geenrl charcterisitic of HPLC/UPLC methods for flavnoid analysis.  

Reviewer 3 Report

Comments and Suggestions for Authors

Not all changes are satisfactory, but I am willing to accept the manuscript in its current form.

Comments on the Quality of English Language

Minor editing of English language required

Author Response

Thank you for your positive comments and valuable suggestions to improve the quality of our manuscript. We have tried our best to polish the language in the revised manuscript.

Reviewer 5 Report

Comments and Suggestions for Authors

My comments are in the attached file.

Comments on the Quality of English Language

Only small mistakes to correct.
